# Wide-ranging consequences of priority effects governed by an overarching factor

**Callie R Chappell[1]\*, Manpreet K Dhami[1,2], Mark C Bitter[1], Lucas Czech[3], Sur Herrera Paredes[1], Fatoumata Binta Barrie[1], Yadira Calderón[1], Katherine Eritano[1], Lexi-Ann Golden[1], Daria Hekmat-Scafe[1], Veronica Hsu[4,5], Clara Kieschnick[1], Shyamala Malladi[1], Nicole Rush[1], Tadashi Fukami[1,6]\***

[1]Department of Biology, Stanford University, Stanford, United States; [2]Biocontrol and Molecular Ecology, Manaaki Whenua - Landcare Research, Lincoln, New Zealand; [3]Department of Plant Biology, Carnegie Institution for Science, Stanford, United States; [4]Department of Ecology, Evolution and Marine Biology, University of California, Santa Barbara, Goleta, United States; [5]Koch Institute for Integrative Cancer Research, Massachusetts Institute of Technology, Cambridge, United States; [6]Department of Earth System Science, Stanford University, Stanford, United States

**\*For correspondence:**
calliech@stanford.edu (CRC);
fukamit@stanford.edu (TF)

**Competing interest:** The authors declare that no competing interests exist.

**Abstract:** Priority effects, where arrival order and initial relative abundance modulate local species interactions, can exert taxonomic, functional, and evolutionary influences on ecological communities by driving them to alternative states. It remains unclear if these wide-ranging consequences of priority effects can be explained systematically by a common underlying factor. Here, we identify such a factor in an empirical system. In a series of field and laboratory studies, we focus on how pH affects nectar-colonizing microbes and their interactions with plants and pollinators. In a field survey, we found that nectar microbial communities in a hummingbird-pollinated shrub, *Diplacus* (formerly *Mimulus*) *aurantiacus*, exhibited abundance patterns indicative of alternative stable states that emerge through domination by either bacteria or yeasts within individual flowers. In addition, nectar pH varied among *D. aurantiacus* flowers in a manner that is consistent with the existence of these alternative stable states. In laboratory experiments, *Acinetobacter nectaris*, the bacterium most commonly found in *D. aurantiacus* nectar, exerted a strongly negative priority effect against *Metschnikowia reukaufii*, the most common nectar-specialist yeast, by reducing nectar pH. This priority effect likely explains the mutually exclusive pattern of dominance found in the field survey. Furthermore, experimental evolution simulating hummingbird-assisted dispersal between flowers revealed that *M. reukaufii* could evolve rapidly to improve resistance against the priority effect if constantly exposed to *A. nectaris*-induced pH reduction. Finally, in a field experiment, we found that low nectar pH could reduce nectar consumption by hummingbirds, suggesting functional consequences of the pH-driven priority effect for plant reproduction. Taken together, these results show that it is possible to identify an overarching factor that governs the eco-evolutionary dynamics of priority effects across multiple levels of biological organization.

## Editor's evaluation

This important study documents the causes and consequences of priority effects in community assembly, using a plant-microbe-pollinator model system. Using an elegant combination of lab and field approaches, the authors provide compelling evidence that early-colonizing microbes alter the

nectar pH, which has far-reaching ecological and evolutionary effects. It will likely be of interest to a wide audience and has implications for microbial, plant, and animal ecology and evolution.

## Introduction

Many ecological communities take one of multiple alternative states even under the same species pool and the same initial environmental conditions (*Beisner et al., 2003*; *Chase, 2003*; *Scheffer et al., 2001*; *Schröder et al., 2005*). Alternative states can vary not just in species composition, but also in functional, ecosystem-level characteristics such as invasion resistance, total biomass, and nutrient cycling (*Bittleston et al., 2020*; *Delory et al., 2019*; *Leopold et al., 2017*; *Pausas and Bond, 2020*; *Sprockett et al., 2018*; *Suding et al., 2004*). However, it is often hard to predict which of the alternative states an ecological community will take. The main challenge is that alternative states are caused by priority effects, the elusive, historically contingent process in which the order and timing of species arrival dictate the trajectory of community assembly (*Debray et al., 2022*; *Drake, 1991*; *Fukami, 2015*; *Palmgren, 1926*; *Song et al., 2021*). To further complicate matters, the strength of priority effects is not always static, but can change rapidly through evolutionary changes in species traits (*De Meester et al., 2016*; *Faillace et al., 2022*; *Faillace and Morin, 2016*; *Knope et al., 2011*; *Urban and De Meester, 2009*; *Wittmann and Fukami, 2018*; *Zee and Fukami, 2018*). The broad scope of priority effects makes it difficult to fully understand which of the alternative states is realized under what conditions (*Fukami, 2015*). For any ecological community, this understanding requires simultaneous examination of the compositional, functional, and evolutionary consequences of priority effects.

Questions that need to be addressed to understand the causes and consequences of priority effects are indeed wide-ranging. For example, how do priority effects happen (mechanism)? When are priority effects particularly strong (condition)? How rapidly can priority effects change in strength (evolution)? And how do priority effects affect the functional, not just taxonomic, properties of the communities being assembled (functional consequences)? These questions are interrelated, but given that they concern different scales of time and biological organization, it seems reasonable to expect each question to involve a different set of species traits and environmental factors. In this paper, we present empirical evidence that, contrary to this expectation, it is possible for one common factor to underlie many aspects of priority effects, including mechanism, condition, evolution, and functional consequences. Specifically, we show that environmental pH is an overarching factor governing priority effects in a microbial system.

Simple microbial systems can help identify general basic principles that organize ecological communities (*Cadotte et al., 2005*; *Drake et al., 1996*; *Jessup et al., 2004*; *Vega and Gore, 2018*), and the nectar microbiome has recently emerged as a well-characterized simple system for understanding community assembly (*Brysch-Herzberg, 2004*; *Chappell and Fukami, 2018*; *de Vega et al., 2021*; *Lachance et al., 2001*; *Letten et al., 2018*; *Vannette, 2020*). Our study system here consists of the bacteria and yeasts that colonize the floral nectar of the sticky monkeyflower, *Diplacus* (formerly *Mimulus*) *aurantiacus*, a hummingbird-pollinated shrub native to California and Oregon of the USA (*Belisle et al., 2012*). Initially sterile, floral nectar is colonized by these microbes via pollinator-mediated dispersal (*Belisle et al., 2012*; *de Vega et al., 2022*). Microbial communities that develop in nectar through this dispersal are often simple, dominated by one or a few species of nectar-specialist bacteria or yeast (*Álvarez-Pérez et al., 2019*; *Golonka and Vilgalys, 2013*; *Herrera et al., 2010*; *Tsuji and Fukami, 2018*; *Warren et al., 2020*).

Our previous work has shown that hummingbird-mediated dispersal is highly stochastic because the pollinators differ in the species identity of yeasts and bacteria they carry on their beaks and tongues (*Morris et al., 2020*; *Toju et al., 2018*; *Vannette et al., 2021*; *Vannette and Fukami, 2017*). Additionally, the outcome of local antagonistic interactions among these microbial species can be sensitive to relative initial abundances (*Dhami et al., 2016*; *Dhami et al., 2018*; *Grainger et al., 2019*; *Mittelbach et al., 2016*; *Peay et al., 2012*; *Tucker and Fukami, 2014*; *Vannette and Fukami, 2014*). Together, the stochastic dispersal and the history-dependent interactions jointly cause priority effects in nectar microbial communities. Moreover, recent studies indicate that whether flowers are dominated by bacteria or yeasts can affect pollination and seed set, although a mechanism for this effect remains unclear (*de Vega et al., 2022*; *Good et al., 2014*; *Herrera et al., 2013*; *Jacquemyn et al.,*

*2021*; *Junker et al., 2014*; *Rering et al., 2020*; *Schaeffer and Irwin, 2014*; *Vannette et al., 2013*; *Vannette and Fukami, 2016*; *Yang et al., 2019*). Despite the increasing amount of knowledge about this system, whether or not there is a common factor that explains the various phenomena associated with priority effects has not been investigated in this system, or any other system for that matter, to our knowledge.

In this paper, we present and synthesize multiple independent pieces of evidence that all point to the pervasive role of pH across several aspects of priority effects in this nectar microbiome. First, we report the results of a field survey of *D. aurantiacus* flowers across a 200-km coastline in California, showing that microbial communities in the nectar exhibit distribution patterns that are consistent with the existence of two alternative community states, one dominated by bacteria and the other dominated by yeasts (*Figure 1A*). Next, we describe findings from laboratory experiments that suggest that the potential alternative states we observed in the field survey are caused by inhibitory priority effects between bacteria and yeasts (*Figure 1B*). We then show that these priority effects are largely driven by bacteria-induced reduction in nectar pH , a finding consistent with the tri-modal distribution of nectar pH among flowers that we observed in the field (*Figure 1A*). Motivated by these results as well as the high variation we found in the strength of priority effects among yeast strains, we then use experimental evolution in artificial nectar (*Figure 1C*) to show that the low pH of nectar can cause rapid evolution of yeast traits, resulting in increased resistance to the pH-driven priority effects (*Figure 1D*). This evolution appears to be constrained by a trade-off between tolerance to a low-pH environment and growth in a neutral-pH environment. Using whole genome resequencing, we explore genomic differences in yeast strains evolved in isolation, in the presence of its primary bacterial competitor, and in a low pH environment (*Figure 1E*). Finally, we show in a field experiment that the low pH of nectar reduces nectar consumption by flower-visiting hummingbirds (*Figure 1F*), suggesting that pH reduction is the likely explanation for our earlier observations that nectar bacteria reduced pollination and seed set in *D. aurantiacus*. We end by weaving these results together in a pH-based story of the wide-ranging consequences of priority effects in this system.

## Results and discussion
### Field observations suggest bacterial vs. yeast dominance as alternative states

To study the distribution of nectar-colonizing bacteria and yeasts among *D. aurantiacus* flowers, we conducted a regional survey in and around the San Francisco Peninsula of California (*Figure 1A*, *Figure 2—source data 1*) in June and July of 2015. Floral nectar was sampled from a total of 1152 flowers (96 flowers at each of 12 sites) along an approximately 200km coastline (*Dhami et al., 2018*; *Tsuji et al., 2016*; *Figure 2A*). Bacteria and yeasts were cultured on different media to characterize floral nectar communities, which had been corroborated by molecular sequencing (*Vannette and Fukami, 2017*) and correlations between cell counts and colony forming units (*Peay et al., 2012*, *Figure 2—figure supplement 1*).

Across all 12 sites, we found that *D. aurantiacus* flowers were frequently dominated by either bacteria or yeast, but rarely by both (*Figure 2C*, *Figure 2—figure supplement 2*). We used a classification method called CLAM (*Chazdon et al., 2011*) to classify flowers into four groups: bacteria-dominated flowers, yeast-dominated flowers, co-dominated flowers, and flowers with too few microbes to be accurately classified (*Figure 2B*, *Figure 2—figure supplement 3*). For each site, we then calculated the proportion of co-dominated flowers that would be expected if bacteria and yeasts were distributed among flowers independently of each other. This analysis showed that the observed proportions of co-dominated flowers were lower than expected by chance alone (*Figure 2D*; paired t-test: n=12, 95% CI [-5.3,–1.1], p=0.006).

This scarcity of co-dominance may have been caused by stochastic factors such as dispersal limitation that creates spatial segregation (*Belisle et al., 2012*) or deterministic factors such as nectar chemistry that allows for niche partitioning. For example, some flowers may have nectar characteristics that intrinsically favor bacterial growth, while other flowers with different nectar chemistry may preferentially support yeast growth. However, another possibility is that the scarcity of co-dominance was caused jointly by stochastic and deterministic factors. Specifically, if a flower happens to become dominated by bacteria, it may prevent yeast from becoming abundant, and vice versa, through

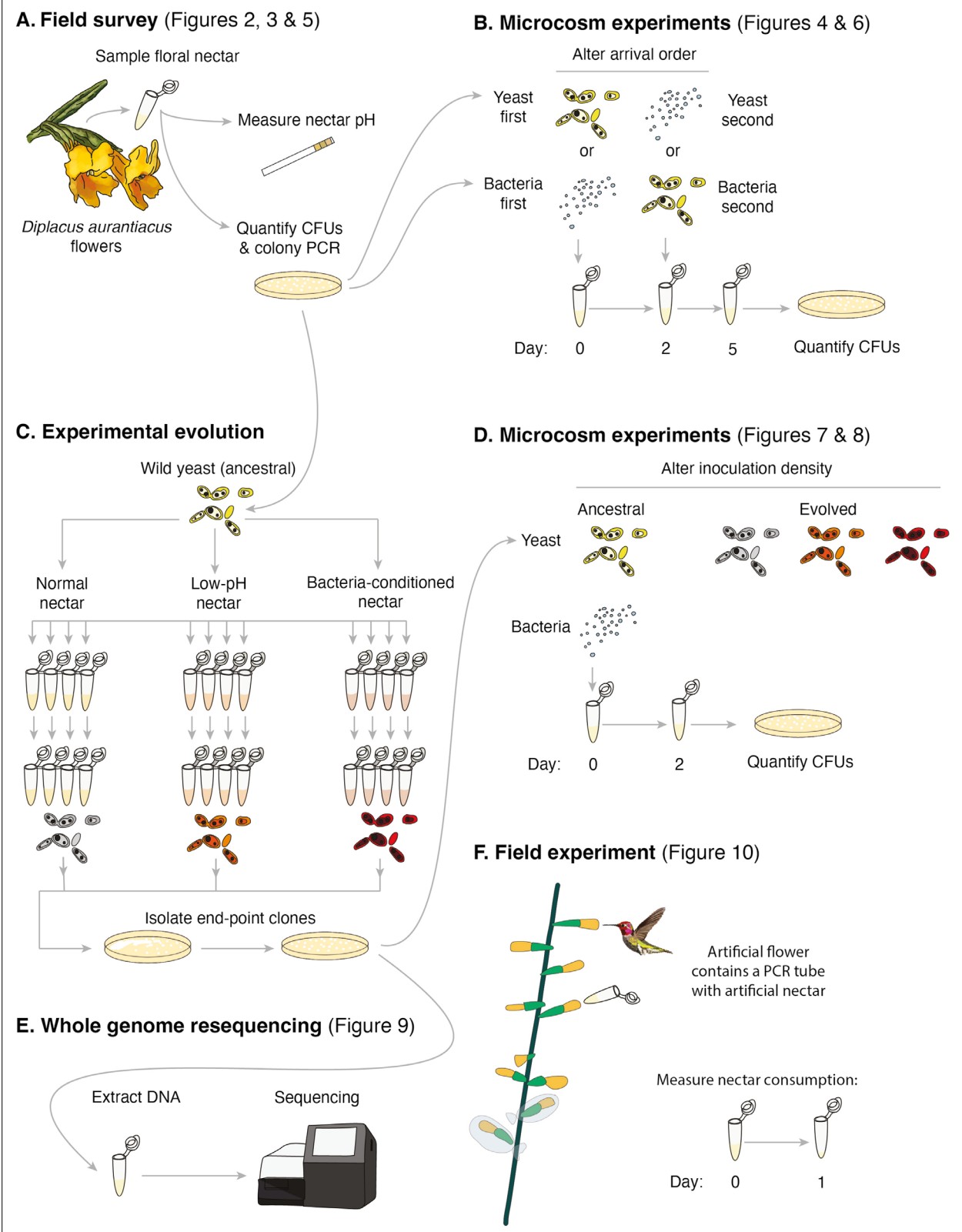

**Figure 1.** Schematic of the approaches taken in this study. (**A**) Field survey to characterize the distribution of yeast and bacteria in flowers as well as nectar pH of *Diplacus aurantiacus*, (**B**) initial microcosm experiments to assess strength of priority effects and identify nectar pH as a potential driver, (**C**) experimental evolution to study adaptation to low-pH and bacteria-conditioned nectar, (**D**) secondary microcosm experiments to study the effect of adaptation to nectar environments, (**E**) whole genome resequencing to identify genomic differences between evolved strains, and (**F**) field experiments to study the effect of low pH on nectar consumption by pollinators.

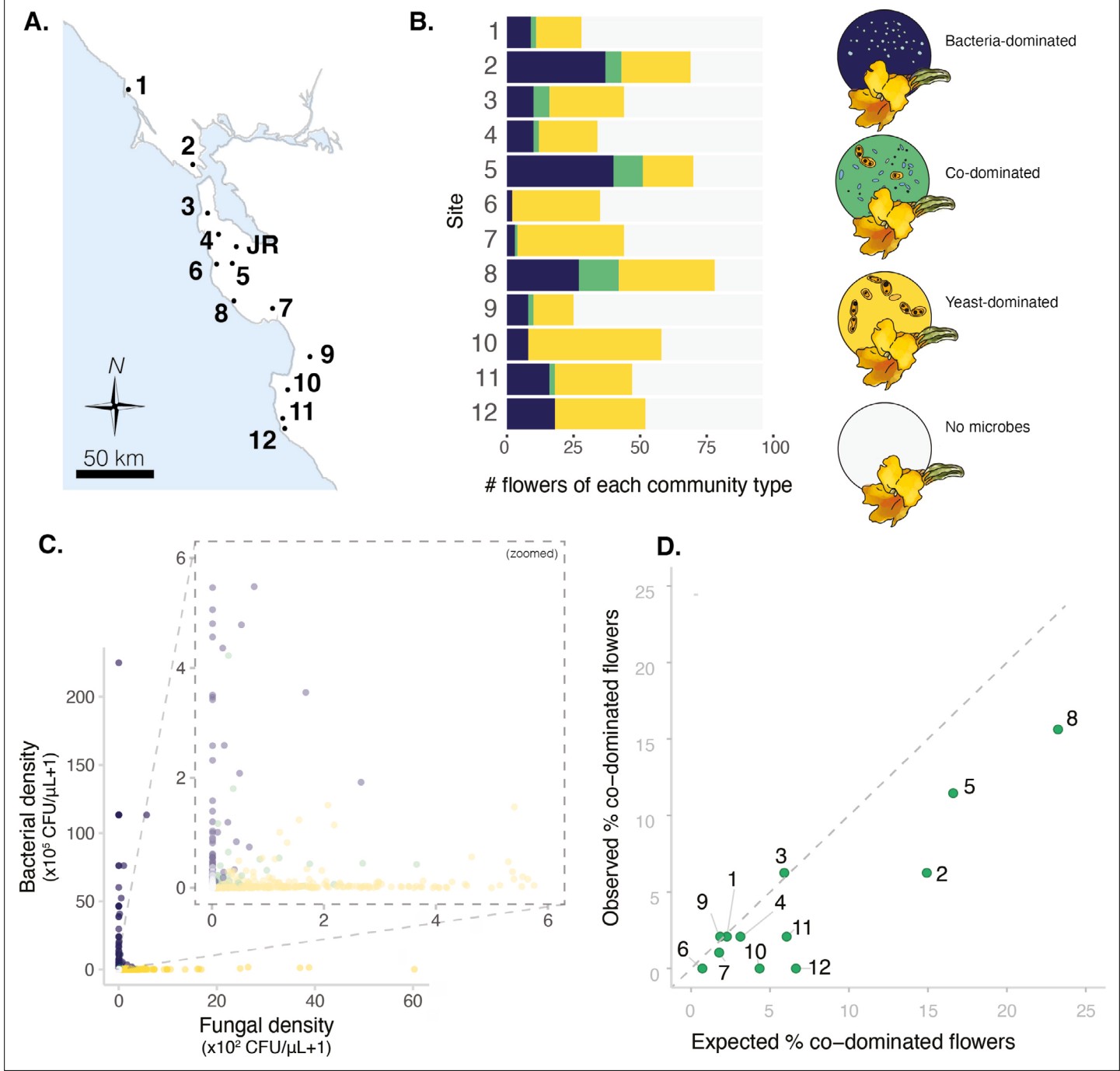

**Figure 2.** Sites vary in regional dominance of bacteria and yeast. (**A**) Ninety-six *Diplacus aurantiacus* flowers were harvested from each of 12 field sites in and around the San Francisco Peninsula in California, USA (***Figure 2—source data 1***) with (**B**) variable numbers of flowers classified as bacteria-dominated (blue), fungi-dominated (yellow), co-dominated (green) flowers, or flowers where microbes were too rare to determine (grey) (n=1152). (**C**) Flowers are often dominated by bacteria or yeast, but rarely both. Each point represents a floral community and inset plot represents zoomed-in version of the plot behind it (n=1152). (**D**) Co-dominated flowers were observed less frequently than expected. In panel D, each point represents a site, with the numbers indicating the site numbers shown in panels A and B. In panel A, the location of Jasper Ridge Biological Preserve (JR) is also indicated (n=12).

The online version of this article includes the following source data and figure supplement(s) for figure 2:

**Source data 1.** Field sites in *Diplacus aurantiacus* field survey.

**Source data 2.** Association between percentage of flowers colonized by yeast or bacteria per plant and the distance between host plants.

*Figure 2 continued on next page*

*Figure 2 continued*

**Source data 3.** Association between bioclimate variables, date of sampling, and microbial colonization.

**Figure supplement 1.** Preliminary association between flow cytometry cell counts (populations identified by forward and side scatter) and colony forming units of *A. nectaris* growing on tryptic soy agar with cycloheximide.

**Figure supplement 2.** *Diplacus aurantiacus* flowers were harvested from 12 field sites in and around the San Francisco Peninsula (California, USA) at various dates in June and July 2015 with variable densities of bacteria and yeast (n=1152).

**Figure supplement 3.** Classification of flowers into bacteria-dominated (blue), yeast-dominated (yellow), co-dominated (green), or microbes too rare to determine (grey).

**Figure supplement 4.** We found no significant relationship between distance and the difference in bacterial colonization (**A**, n=8775, p=0.07, $R^2$=0.0003) and a slightly negative association between distance and the difference in fungal colonization (**B**, n=8775, p<0.05, $R^2$=0.004).

differential modification on nectar chemistry by bacteria vs. yeasts. This mutual suppression would represent inhibitory priority effects, where stochastic dispersal dictates the trajectory of local community assembly because early-arriving colonists deterministically exclude late-arriving immigrants.

To examine the possibility of spatial segregation, we regressed the geographic distance between all possible pairs of plants to the difference in bacterial or fungal abundance between the paired plants. If plant location affected bacterial or yeast abundance, we should see a positive relationship between distance and the difference in abundance between a given pair of plants. Contrary to this expectation, we found no strong relationship between distance and the difference in bacterial colonization (*Figure 2—figure supplement 4A*, *Figure 2—source data 2*, LM: n=8775, p=0.07, $R^2$=0.0003) and a slightly negative association between distance and the difference in fungal colonization (*Figure 2— figure supplement 4B*, *Figure 2—source data 2*, LM: n=8775, p<0.05, $R^2$=0.004), suggesting that spatial segregation is unlikely to explain the observed abundance pattern.

It is also possible that climatic variables affected the colonization of bacteria and yeasts. However, in a linear mixed model predicting bacterial or yeast abundance by average annual temperature, temperature seasonality, annual precipitation, sampling date, and site location included as a random effect, none of the predictors were significant (*Figure 2—source data 3*). This result indicates that the observed abundance pattern is unlikely to have been strongly influenced by spatial proximity, temperature, moisture, or seasonality, reinforcing the hypothesis that the distribution pattern is instead underlain by bacterial and yeast dominance as alternative stable states. To test this hypothesis more directly, we conducted laboratory experiments using *Acinetobacter nectaris* and *Metschnikowia reukaufii*, which are the most frequently found species of nectar bacteria and yeasts, respectively.

Previously, we reported that *M. reukaufii* was the most frequently cultured species of fungi in *D. aurantiacus* nectar in the 12-site survey in 2015 (Table S2 in *Dhami et al., 2018*; see also *Belisle et al., 2012*). As for bacteria, Dhami et al.'s (2018) data are not extensive enough to draw a firm conclusion. However, we here present data from a multi-year survey of bacteria cultured from *D. aurantiacus* nectar at one site, Jasper Ridge (JR, *Figure 3*), from 2012 to 2022. The Jasper Ridge data support culture-independent (metabarcoding) data on bacterial species composition of *D. aurantiacus* nectar at the same site (*Vannette and Fukami, 2017*) and a nearby site (*Toju et al., 2018*). Namely, both culture-dependent and culture-independent methods indicate that *Acinetobacter* spp. were the dominant species of bacteria in *D. aurantiacus* nectar, followed by *Neokomagataea* (formerly *Gluconobacter*) sp. (see further detail in Appendix 1). The most common species of bacteria and yeast observed in our study system, such as species of *Acinetobacter*, *Neokamagataea*, *Pseudomonas*, and *Metschnikowia*, have been shown to be common nectar specialists in other plants as well (e.g., *Fridman et al., 2012*; *Warren et al., 2020*). Taken together, these independent pieces of information collectively indicate that, given their prevalence, *A. nectaris* and *M. reukaufii* would be a reasonable pair of species to focus on as a first step toward understanding the possible alternative states in the nectar microbial community of *D. aurantiacus* in our study landscape.

## Evidence for pH-mediated priority effects causing alternative states

One way to directly test for priority effects between bacteria and yeasts is to experimentally manipulate initial relative abundance of the two microbial groups. This manipulation would allow us to determine if communities diverge such that bacteria-dominated communities develop if bacteria were

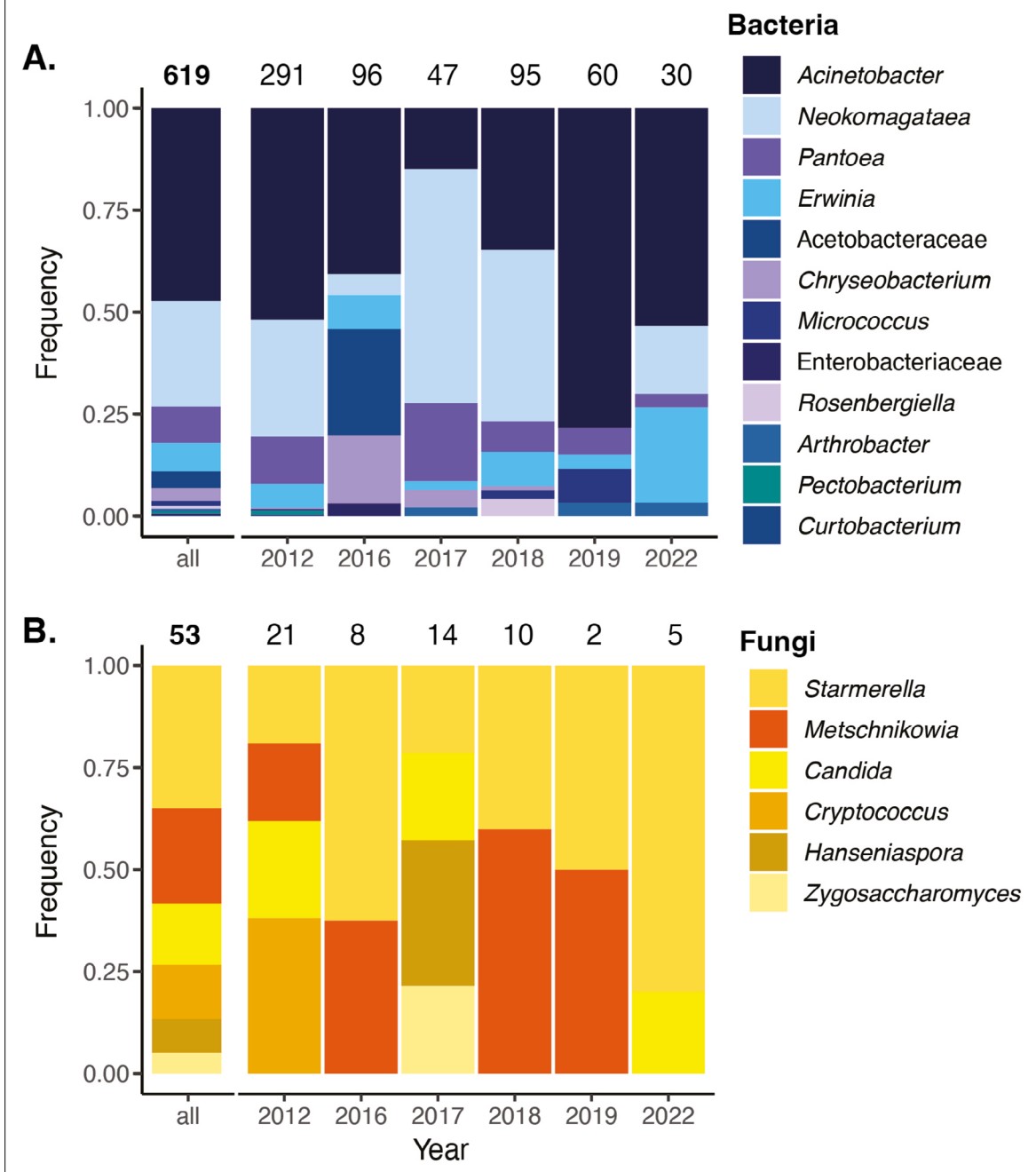

**Figure 3.** Cultured bacteria and yeast from a 6year survey. Cultured nectar bacteria (**A**) and yeast (**B**) from a 6-year survey of *D. aurantiacus* nectar at Jasper Ridge identified by colony PCR. The number placed at the top of each bar indicates the number of colony samples analyzed. Single fungal colonies were isolated on yeast mold agar (YMA) with supplemented 100 mg/L of the antibacterial chloramphenicol. Single bacterial colonies were either isolated on Reasoner's 2A agar (R2A) supplemented with 20% sucrose and 100 mg/L of the antifungal cycloheximide (2012–2018), or tryptic soy agar (TSA) supplemented with 100 mg/L of the antifungal cycloheximide (2019–2022).

The online version of this article includes the following source data for figure 3:

**Source data 1.** Primer sequences and PCR cycles for colony PCR.

initially more abundant than yeast and vice versa. Our prior work that took this approach yielded evidence for strong priority effects in *D. aurantiacus* nectar (**Dhami et al., 2016**; **Peay et al., 2012**; **Toju et al., 2018**; **Vannette and Fukami, 2014**). However, these studies either looked at priority effects among yeast species or, in the case of priority effects between yeast and bacteria, focused on the yeast *M. reukaufii* and the bacterium *Neokomagataea* (formerly *Gluconobacter*) sp. (**Tucker and**

*Fukami, 2014*). The most common bacterial species, *A. nectaris*, may also engage in strong priority effects against *M. reukaufii*, but this possibility has not previously been tested.

To test for priority effects between *M. reukaufii* and *A. nectaris*, we used sterile PCR tubes as artificial flowers. Each tube contained artificial nectar that closely mimicked sugar and amino acid concentrations in field-collected *D. aurantiacus* nectar (*Peay et al., 2012*). We altered the arrival order of *M. reukaufii* and *A. nectaris* and measured growth after five days, which approximated the lifespan of individual *D. aurantiacus* flowers (*Peay et al., 2012*; *Figure 4—source data 1*). We found that *A. nectaris* exerted a strong inhibitory priority effect against *M. reukaufii* (*Figure 4A*, *Figure 4—source data 2*) and vice versa (*Figure 4—figure supplement 1*).

From our previous work with *Neokomagataea* (*Tucker and Fukami, 2014*), we hypothesized that inhibitory priority effects against yeast were caused by bacteria-induced reduction in nectar pH, hindering yeast growth. Consistent with this hypothesis, we found that higher densities of bacteria were tightly associated with lower nectar pH (Spearman rank-order: n=94, $R^2$(adj.)=0.69, p=2.2 × $10^{-16}$, *Figure 4B*), whereas higher densities of yeast were weakly associated with higher nectar pH (*Figure 4—figure supplement 2*). In addition, monoculture yeast grew poorly in low-pH nectar (pH = 3) when inoculated at low density (*Figure 4C*), but not if introduced at high density (*Figure 4—figure supplement 3*). Bacterial growth was not affected by lowered nectar pH (*Figure 4—figure supplement 4*). Together, these data provide further evidence that nectar pH reduction by bacteria mediates their priority effects on yeast.

One likely reason for this density-dependent effect of pH on yeast is that, when yeast arrive first to nectar, they deplete nutrients such as amino acids and consequently limit bacterial growth, thereby avoiding pH-driven suppression that would happen if bacteria were initially abundant (*Tucker and Fukami, 2014*; *Vannette and Fukami, 2018*). Another possible reason is that larger yeast populations have higher phenotypic heterogeneity, which increases the probability that the population will include cells that are resilient to low pH. This mechanism has been suggested in the baker's yeast, *Saccharomyces cerevisiae* (*Guo and Olsson, 2016*), although whether it applies to *M. reukaufii* remains to be tested. In addition to priority effects, phenotypic plasticity or stochastic phenotypic switching could contribute to increased variation in the extent of bacterial dominance among flowers (*Levy, 2016*; *Morawska et al., 2022*). In fact, previous work on *M. reukaufii* suggested that epigenetic changes contribute to its broad niche width (*Herrera et al., 2010*). However, our experiments manipulating initial abundance provide evidence for priority effects between nectar bacteria and yeast. Since we have already sought to elucidate priority effects of early-arriving yeast in our previous papers (*Tucker and Fukami, 2014*; *Vannette and Fukami, 2018*), we focus primarily on the other side of the priority effects, where initial dominance of bacteria inhibits yeast growth, in this paper.

To assess whether the nectar pH reduction by bacteria that we observed in the laboratory experiment (*Figure 4B*) had relevance for explaining variation in nectar pH among real flowers in the field, we conducted a survey of *D. aurantiacus* nectar at two sites, San Gregorio (site 6) and Jasper Ridge, in June and July of 2022. We found that the distribution of nectar pH ranged from 2 to 9 in a way consistent with the prediction that fresh *D. aurantiacus* nectar has a mean pH of about 7.5 and that, once colonized, yeast and bacteria reduce nectar pH to an average of 5.5 and 2.5, respectively (*Vannette et al., 2013*). Specifically, we found that the distribution of nectar pH in the field was non-unimodal (Hartigan's dip test: n=576, D=0.05, p<2.2 × $10^{-16}$). Furthermore, a 3-mode model was a better fit than a 2-mode model (Likelihood ratio test: p=1.6 × $10^{-16}$, AIC$_{k=2}$ = 2365, AIC$_{k=3=}$2296, *Figure 5—figure supplement 1*). According to the mixture model we used, nectar pH had local modes at 7.8, 5.5, and 2.6 (*Figure 5A*, solid vertical lines), which are strikingly similar to those from the experiment we reported previously (*Vannette et al., 2013*), where filtered nectar from newly opened *D. aurantiacus* flowers was inoculated with no microbes, the yeast *M. reukaufii*, or the bacterium *Neokomagataea* sp. under controlled laboratory conditions (*Figure 5A*, dashed vertical lines). In addition, we observed that older and pollinated flowers were more likely to have low nectar pH, reflecting the fact that, in these flowers, microbes would have more chance to grow and modify the nectar pH than in younger and non-pollinated flowers (*Figure 5B*, *Figure 5—figure supplement 2*). We also found that the two sites differed in their distributions of nectar pH (*Figure 5B*, *Figure 5—figure supplement 2*). At Jasper Ridge, many of the pollinated flowers had low pH values consistent with bacterial dominance. In contrast, at San Gregorio, these flowers had intermediate pH values that characterize yeast dominance. At another site within the region, Stanford University's Dish hill, we measured both nectar pH

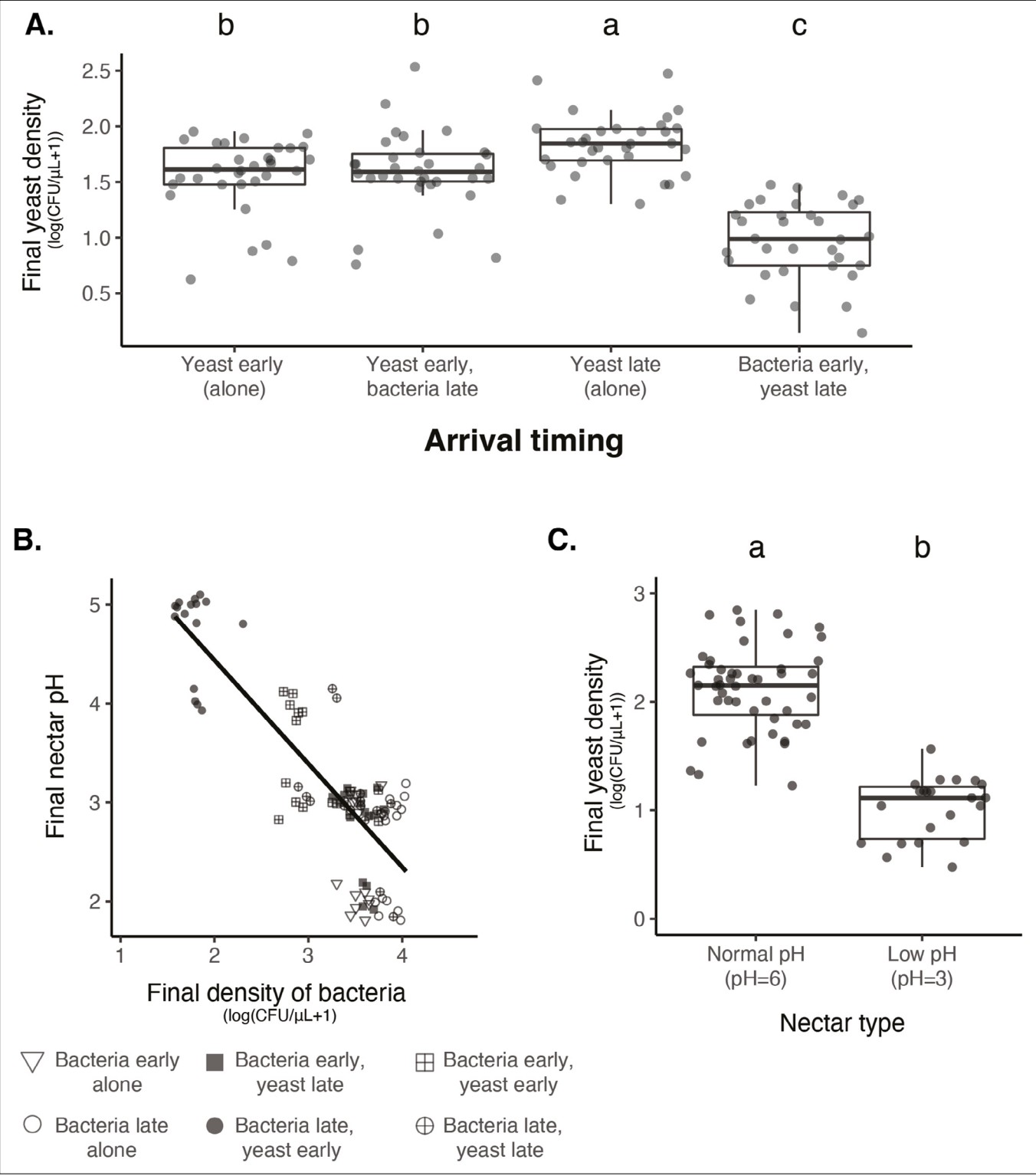

**Figure 4.** Nectar bacteria exert negative priority effects against nectar yeast, potentially due to reduction in nectar pH. (**A**) *Metschnikowia reukaufii* (strain MR1) yeast population density after five days of growth with alternating arrival order with *Acinetobacter nectaris* bacteria or growth alone with inoculation on either the first or third day (arriving early or late) of the experiment (n=128). (**B**) Final pH of nectar after 5 days of bacterial growth; higher densities of bacteria are associated with lower pH. The shape of each point represents the treatments represented in panel A. Points are jittered on the y-axis (n=96). (**C**) Low pH (pH = 3) nectar depresses yeast growth when grown in low-density monoculture (n=72). In box plots (panels A and C), treatments that share the same letter placed above their boxes were statistically indistinguishable from one another.

*Figure 4 continued on next page*

*Figure 4 continued*

The online version of this article includes the following source data and figure supplement(s) for figure 4:

**Source data 1.** Priority effect experiment treatments.

**Source data 2.** Priority effect experiment results.

**Figure supplement 1.** *M. reukaufii* yeast and *A. nectaris* bacteria exhibit negative priority effects against each other, as evidenced by growth in microcosm experiments where arrival order is altered.

**Figure supplement 2.** Yeast increases nectar pH (p<0.05, Spearman rank correlation).

**Figure supplement 3.** We found no effect of nectar type (pH=3, pH=6) on the growth of *M. reukaufii*, when grown in monoculture at a high density (approximately 10,000 cells/μL).

**Figure supplement 4.** We found no effect of nectar type (pH=3, pH=6) on the growth of *A. nectaris* when grown in monoculture at a low density (approximately 10 cells/μL) (**A**) or high density (approximately 10,000 cells/μL) (**B**).

and bacterial growth and found that *D. aurantiacus* flowers with higher bacterial densities tended to have lower nectar pH (LMM: n=62, p=7.4 × 10$^{-8}$) (*Figure 5C*). All of these results give field-based support for the idea that microbial abundance and nectar pH are dictated by the priority effects between bacteria and yeasts.

## pH as an eco-evolutionary driver of priority effects

In the field survey of 12 sites, we found large site-to-site variation in the relative prevalence of bacteria vs. yeast (*Figure 2B*), even though all sites followed the same L-shaped pattern that characterized alternative bacteria vs. yeast dominance (*Figure 2C*). For example, San Gregorio (site 6) had many bacteria-dominated flowers in 2015, whereas a nearby site, La Honda (site 5), had more yeast-dominated flowers that same year. Through further laboratory experimentation, we also found that three genetically divergent strains of *M. reukaufii*, all of which were collected from the same site, Jasper Ridge, were differentially affected by early arrival of *A. nectaris* (*Figure 6*, *Figure 6—source data 1*). Specifically, one of the three yeast strains, MR1, was more negatively affected by priority effects than the other two strains, MY0182 and MY0202, which were representative of distinct genotypic groups found across the 12 sites (*Dhami et al., 2018*). Given this genotypic and phenotypic variation among strains, it seemed conceivable that the site-to-site differences in the relative prevalence of yeasts and bacteria observed in the field could drive an eco-evolutionary feedback (sensu *Post and Palkovacs, 2009*), such that sites dominated by bacteria would be characterized by stronger selective pressure for yeasts to become more resistant to priority effects than sites dominated by yeasts (*Figure 2B*).

To test this hypothesis, we experimentally evolved nectar yeast under conditions that resembled repeated exposure to bacterial priority effects (*Figure 1C*). In our experimental evolution, we simulated the serial transfer of microbes between flowers via hummingbirds over the course of an entire flowering season. To this end, over 60 days, we serially transferred a single clone of *M. reukaufii* strain MR1, which we refer to as ancestral, to fresh nectar three times a week (Appendix 2, *Appendix 2—figure 1*). This strain was isolated from Jasper Ridge (*Peay et al., 2012*). Sixty days roughly corresponds to the duration of a typical flowering season of *D. aurantiacus* at that site (*Belisle et al., 2012*) and some of the other sites we used for the field survey (*Figure 2A*). We chose strain MR1 for this experiment because it was the most susceptible to priority effects (*Figure 6*).

We serially transferred four independent replicates of strain MR1 in three nectar types: (1) synthetic nectar where pH was 6, simulating fresh *D. aurantiacus* nectar (*Vannette et al., 2013*); (2) synthetic nectar where pH was 3 (called "low-pH"), mimicking the pH reduction caused by growth of *A. nectaris*, our hypothesized mechanism of priority effects (*Figure 4—figure supplement 2*); and (3) synthetic nectar where pH was initially 6, but then conditioned by growth of *A. nectaris*. The growth of *A. nectaris* reduced pH to approximately 3 (called 'bacteria-conditioned') (*Figure 4B*). To prepare the bacteria-conditioned nectar, we grew a strain of *A. nectaris* for five days in fresh synthetic nectar and then filtered the bacteria out with a 0.2μm filter. We selected these two experimental treatments to test whether pH reduction alone or, additionally, the production or consumption of chemical compounds by *A. nectaris* influenced *M. reukaufii*'s susceptibility to priority effects. Together, these nectar treatments represent the most extreme selective pressure that yeasts could be exposed to in

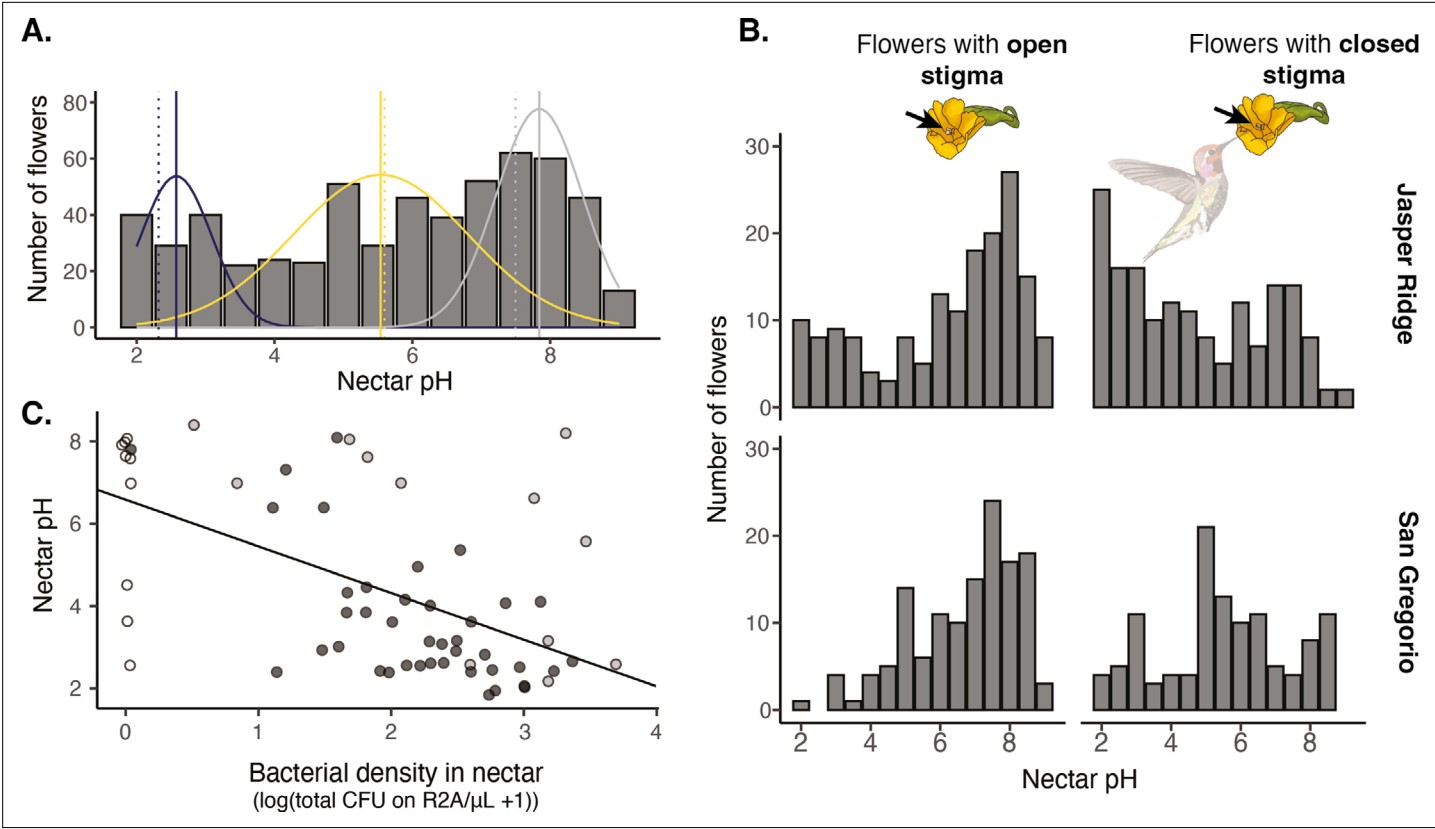

**Figure 5.** Field survey of nectar pH in *D. aurantiacus*. (**A**) Distribution of nectar pH in individual *D. aurantiacus* flowers collected at Jasper Ridge and San Gregorio (n=576 flowers from which we were able to extract a sufficient amount of nectar to measure pH; 21% of flowers sampled, i.e.,152 of 728 flowers, had too little nectar for pH measurement). The tri-modal distributions represent the prediction from a 3-part mixture model, with the modes indicated by solid vertical lines. Dashed vertical lines indicate experimental pH measurements from *Vannette et al., 2013*, where bacteria (blue dashed vertical line) and yeast (yellow dashed vertical line) were grown in field-collected *D. aurantiacus* nectar (control as grey dashed vertical line), and pH was measured after four days of growth. (**B**) Distributions of nectar pH among *D. aurantiacus* flowers with open and closed stigmas (a closed stigma indicates visitation by a pollinator in *D. aurantiacus*; *Fetscher and Kohn, 1999*) shown separately for Jasper Ridge and San Gregorio (site 6). At Jasper Ridge, 71/233 and 37/204 flowers with closed and open stigma, respectively, had too little nectar to measure pH. At San Gregorio, 31/145 and 13/146 flowers with closed and open stigma, respectively, had too little nectar to measure pH. (**C**) Association between bacterial density in individual flowers and nectar pH (n=62). White points represent flowers where no microbes were cultured on R2A, but some colonies were present on TSA. Grey points represent flowers where yellow colonies were present on R2A at a greater density than other colonies on R2A. Preliminary data suggest that these yellow colonies represent non-acidifying bacteria such as *Pseudomonas*. Black points represent flowers with colonies on R2A that do not fit into either of the prior categories. The regression line was shown for all data.

The online version of this article includes the following figure supplement(s) for figure 5:

**Figure supplement 1.** Expectation-maximization (EM) algorithm-based mixture model of nectar pH from *D.aurantiacus* flowers harvested from Jasper Ridge and San Gregorio in June-July 2022.

**Figure supplement 2.** Distribution of nectar pH for flowers harvested from Jasper Ridge and San Gregorio separated by anther status.

the field, either never encountering flowers with early-arriving bacteria or only encountering flowers where bacteria had already colonized.

To determine whether yeasts that had evolved in low-pH nectar were more resistant to priority effects by bacteria, we conducted laboratory experiments similar to those described above. We manipulated the initial density of a given *M. reukaufii* strain relative to a co-inoculated standard strain of *A. nectaris* and measured their abundances after three days of growth. Using these data, we quantified the strength of the bacterial priority effect on yeast using the same metric as before (*Figure 7*).

We found that, over the two months of serial transfers, yeasts that evolved in environments that constantly simulated early arrival of bacteria (bacteria-conditioned and low-pH nectars) were less affected by priority effects than the ancestral yeast or the yeast evolved in the normal nectar (*Figure 7A*, *Figure 7—figure supplement 1*, *Figure 7—source data 2*). This difference in the

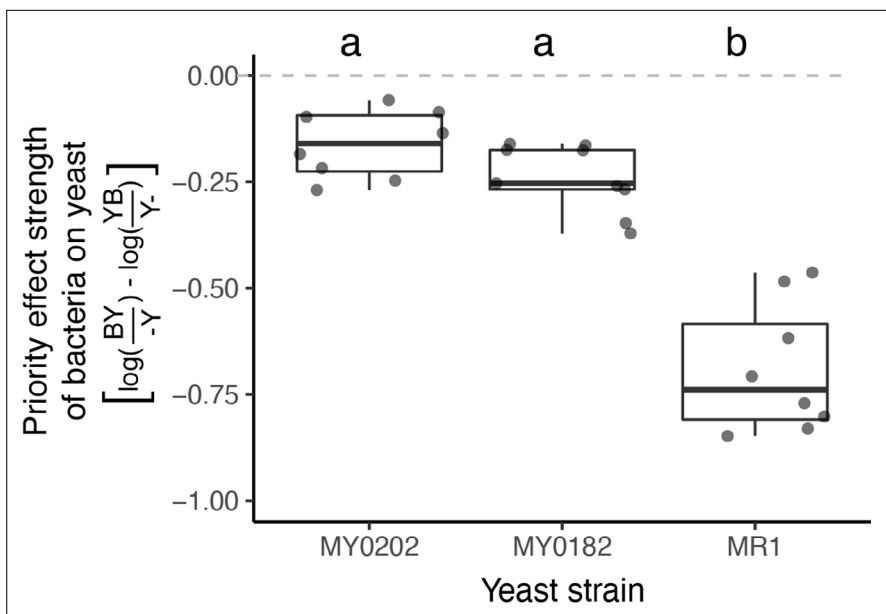

**Figure 6.** *M. reukaufii* strains differ in susceptibility to bacterial priority effects. Three strains of *M. reukaufii* were differentially affected by early arrival of bacteria (n=25). For each strain, we calculated the strength of priority effects using a metric that compares growth between varying initial inoculation densities with a competitor and alone. BY and YB represents initial dominance by bacteria or yeast (e.g. BY represents bacteria arriving early and yeast arriving late). Y- and -Y represent the comparable growth of yeast at either density (early or late) alone. Letters shown above each box (each treatment) indicate statistical significance as in *Figure 4*.

The online version of this article includes the following source data for figure 6:

**Source data 1.** Priority effect experiment results with wild strains.

strength of priority effects reflects that the absolute growth of yeast evolved in bacteria-conditioned and low-pH nectars was less negatively affected by bacteria (*Figure 7B*, *Figure 7—source data 3*). However, their improved growth was apparent only when bacteria were present (*Figure 7—figure supplement 2*, *Figure 7—source data 4*). Together, these findings suggest that yeast can evolve to resist low pH, the very mechanism by which bacteria exert priority effects against yeast over a duration that roughly corresponds to a single flowering season of *D. aurantiacus*.

Furthermore, we observed a pattern that suggests a potential evolutionary tradeoff between adaptation to low pH and neutral pH (*Figure 8*, *Figure 8—source data 1*). When considered across all experimental evolution treatments, strains that evolved stronger resistance to priority effects of pH-reducing bacteria became marginally worse at growing in neutral-pH monoculture (strains that are in the upper left quadrant in *Figure 8*). Conversely, strains that did not increase resistance to bacterial priority effects during experimental evolution were better able to grow in neutral-pH monoculture (those in the upper right quadrant in monoculture) (LMM: n=48, p=0.05, *Figure 8—figure supplement 1*). Incidentally, this type of evolutionary trade-off is the basis for the 'eco-evolutionary buffering' hypothesis recently proposed by *Wittmann and Fukami, 2018*. This hypothesis provides one potential reason why species that engage in strong inhibitory priority effects in local communities can still co-exist in a metacommunity even though such priority effects should eventually cause only one species to persist in the metacommunity, with others driven to extinction. Our finding here, that is rapid evolution of resistance to priority effects under a trade-off constraint, may partly explain why both yeast and bacteria persist in the field despite strong priority effects, but this possibility remains highly speculative at this stage.

## Genome re-sequencing to explore the genetic changes associated with pH-driven rapid evolution

To understand whether these phenotypic differences in response to priority effects have a genetic basis, we conducted full genome re-sequencing of the ancestral strain and end-point clones isolated

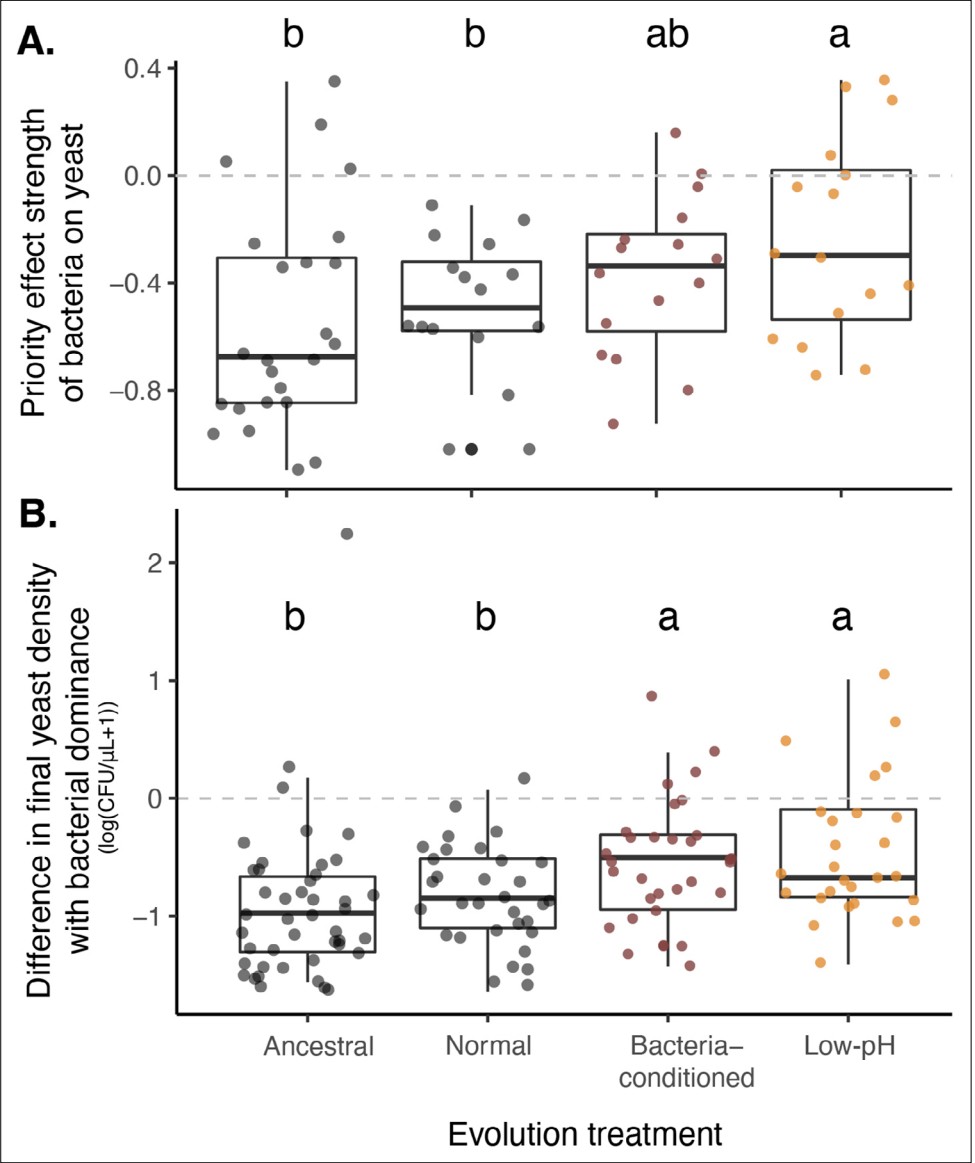

**Figure 7.** Yeast evolved with bacterial niche modification were more resistant to bacterial priority effects. (**A**) Yeast evolved in low-pH nectar was less affected by bacterial priority effects than other treatments, especially compared to ancestral yeast and yeast evolved in normal nectar (n=72). Consequently, (**B**) yeast evolved in bacteria-like nectar (both bacteria-conditioned and low-pH) was less negatively affected by initial bacterial dominance, relative to their growth alone, than ancestral yeast or yeast evolved in normal nectar (n=168). Letters shown above each box (each treatment) indicate statistical significance as in *Figure 4*.

The online version of this article includes the following source data and figure supplement(s) for figure 7:

**Source data 1.** Evolution treatments.

**Source data 2.** Priority effect experiment results with evolved strains (priority effect metric).

**Source data 3.** Differences in growth between evolved strains with and without bacteria.

**Source data 4.** Difference in final yeast densities between evolved strains with and without bacteria.

**Figure supplement 1.** We calculated an additional priority effect metric, which corroborated our main result.

**Figure supplement 2.** Effects of evolution treatments and bacterial addition on yeast density.

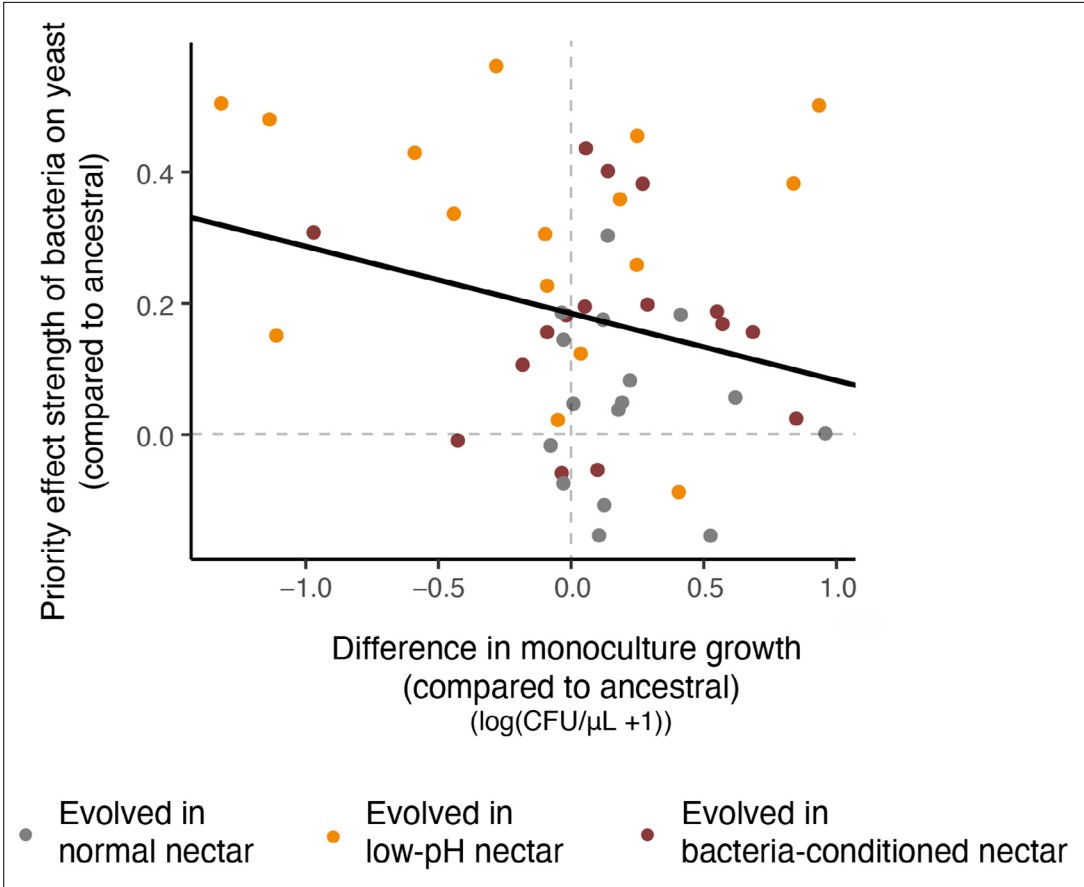

**Figure 8.** Relationship between resistance to priority effects and monoculture growth. Strains that could more strongly resist priority effects by bacteria were poor at growing in monoculture (upper left quadrant). Conversely, strains that were more affected by bacterial priority effects were better able to grow in monoculture (upper right quadrant). Each point in the plot represents an evolved strain, plotted with respect to the ancestral strain grown in that round of the experiment (centered on the origin) (n=60).

The online version of this article includes the following source data and figure supplement(s) for figure 8:

**Source data 1.** Relationship between resistance to priority effects and monoculture growth.

**Figure supplement 1.** Negative relationship between the strength of priority effects of bacteria and yeast on their monoculture growth rate at a low density, with round of the experiment (week) as a random effect (LMM: n=48, p=0.006).

from the four independent evolutionary replicates of each treatment (*Figure 1E*, *Figure 9—figure supplement 1* , *Figure 9—source data 1*). Genomic variation across samples was characterized by calling variants against the reference genome for this species (*Dhami et al., 2016*). Our sequencing approach discounted the possibility that the observed phenotypic dynamics were solely driven by plasticity or epigenetic changes in the yeast strains (*Herrera et al., 2012*). We explored two potential mechanisms of adaptation: loss of heterozygosity through fixation of one of the ancestral heterozygous alleles and de novo mutations that emerge and become fixed.

Loss of heterozygosity (LOH) is a common driver of molecular evolution in yeast (*Dutta et al., 2021*; *Peter et al., 2018*). We explored its role in driving the observed patterns of evolution between the ancestral strain and our evolved lineages. We found that the ancestral strain exhibited heterozygosity at 25,820 sites across the 19 Mb diploid genome. Across all evolved lineages, we observed an average of 142 sites exhibiting LOH per lineage (1699 total across all lineages). Most sites exhibiting LOH were shared across lineages and treatments (97%) and likely indicative of adaptation of this wild strain to the laboratory environment.

To infer genomic loci subject to treatment-specific selection pressure, we compared patterns of LOH between the lineages evolved in the normal nectar treatment and those evolved in the low-pH

and bacteria-conditioned treatments. We specifically identified individual sites of LOH that were shared across multiple lineages within the low-pH and/or bacteria-conditioned treatments, but were rare or absent in the normal treatment and ancestral strain (*Figure 9—figure supplement 2*). These sites correspond to permutation-derived p-values less than 0.1 and $F_{ST}$ values ranging between 0.3 and 0.5, and thus are indicative of substantial genomic differentiation between treatments.

Despite the limited statistical power owing to the moderate sample size (four evolutionary replicates per treatment), we were able to identify several sites with consistent loss of heterozygosity across the bacteria-conditioned and low-pH treatments (*Figure 9A*). Several of the LOH events most predominant in the low-pH and bacteria-conditioned treatments fell near genes involved in amino acid biosynthesis (*Figure 9—figure supplement 2*, *Figure 9—source data 2* for all treatments). Competition for amino acids has been suggested as a mechanism by which yeast species exert priority effects against bacteria and other species of yeast in this system (*Dhami et al., 2016*; *Peay et al., 2012*; *Tucker and Fukami, 2014*). Our results suggest that adaptation to limited amino acid availability could be associated with differential resistance to priority effects in low-pH environments, though further experimentation is needed to determine the precise targets of selection.

We identified 146 putative de novo mutations across all lineages, with an average of 12 putative de novo mutations per lineage (*Figure 9B*). The locations of these mutations displayed some spatial concordance with patterns of LOH, corroborating the putative role of these genomic regions in adaptive evolution (*Figure 9—figure supplement 2*). Yeast evolved in low-pH and bacteria-conditioned nectar both exhibited putative de novo mutations near genes involved in solute regulation, such as membrane transport, vesicle cargo recognition, and solute carrier proteins (*Figure 9—source data 3*). Solute regulation has been implicated in adaptation to high-osmolarity environments (*Pozo et al., 2015*), where osmotolerant yeast has been found to compete more strongly in sugar-rich nectar. Although we cannot yet directly establish that these are causal variants, they are consistent with our hypothesis that yeasts undergo rapid genetic adaptation to pH-mediated priority effects in nectar.

We also found some differences between yeast evolved in low-pH and bacteria-conditioned nectars. For example, strains evolved in low-pH nectar had repeated LOH events in genes associated with stress responses and growth, whereas strains evolved in bacteria-conditioned nectar did not. Further studies are necessary to deduce the effect that such genomic differences have on their phenotypes and strategies to resist priority effects. Although our phenotypic data suggest that alterations to pH drive priority effects, yeast may take a variety of molecular approaches to resist priority effects by bacteria. Additionally, these results suggest that yeast might develop multiple genomic strategies to resist priority effects, potentially explaining the genetic diversity we see in the field (*Dhami et al., 2018*).

In summary, these results indicate that the observed patterns of phenotypic evolution were concurrent with changes in genomic variation. Future work within this system could elucidate the specific loci underpinning yeast adaptation to bacterial priority effects. These results also highlight that, although our phenotypic data suggest alterations to pH as the most important factor for this priority effect, it still may be one of many affecting the coevolutionary dynamics of wild yeast in the microbial communities they are part of. In the full community context in which these microbes grow in the field, multi-species interactions, environmental microclimates, etc. likely also play a role in rapid adaptation of these microbes, which was not investigated in the current study.

## Reduced consumption of low-pH nectar by flower-visiting animals

Finally, we present evidence that bacteria-induced reduction in nectar pH not only drives microbial priority effects, but also has functional consequences for *D. aurantiacus* plants. Prior work has shown that bacterial growth in nectar can negatively affect nectar consumption by pollinators, the probability of pollination, and the number of seeds produced by *D. aurantiacus* flowers (*Vannette et al., 2013*; *Vannette and Fukami, 2016*). However, the mechanism behind this effect remains unknown. To test whether pH affects nectar consumption by flower-visiting animals, we conducted a field experiment using an array of artificial plants at an outdoor garden at the Stock Farm Growth Facility on the Stanford University campus, located 5 km from Jasper Ridge. We used artificial flowers attached to garden stakes (*Figure 10A*) and measured changes in nectar volume after one day, correcting for reduction in volume through evaporation as opposed to consumption by animals like hummingbirds and bees.

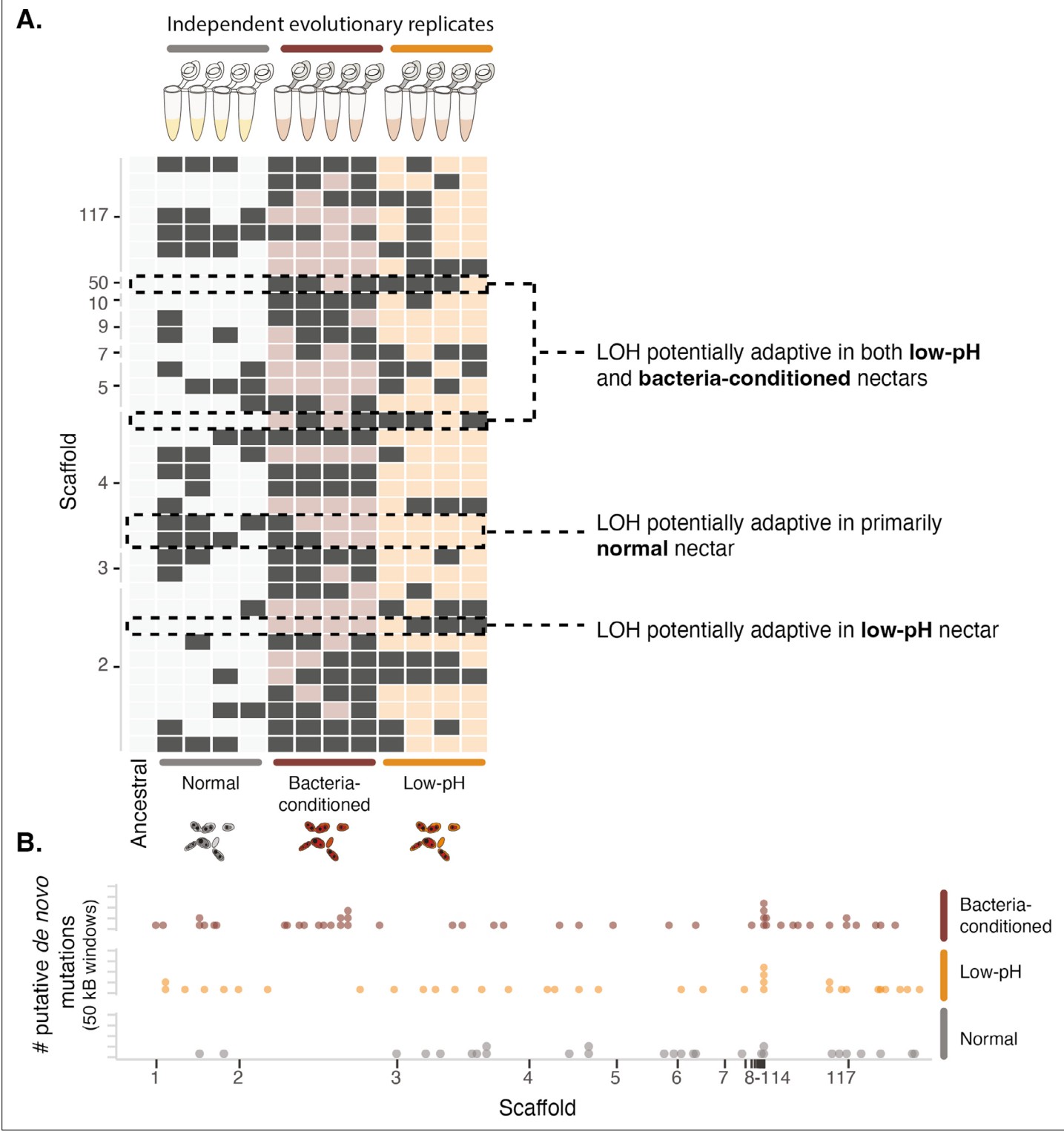

**Figure 9.** Yeast evolved to synthetic nectar. (**A**) Heat map depicting treatment-specific loss of heterozygosity (LOH). Columns represent individual samples (independent evolutionary trajectories) and rows represent single sites with LOH. White or light red/orange boxes indicate sites without LOH, while dark grey boxes represent a site with LOH. Sites selected for the figure exhibited $F_{ST}$ >0.3 and permutation-derived p-value <0.1 when comparing the ancestral strain and one of the evolved clones at that site. Boxes with dashed lines are highlighted examples of sites that are potentially adaptive in low-pH nectar, both low-pH and bacteria-conditioned nectar, and normal nectar but not the other two nectar types. (**B**) Distribution of putative de novo mutations across the genome in 50 kB windows and separated by treatment. Dots represent the sum of putative de novo mutations in a 50 kB window, per treatment (n=146).

*Figure 9 continued on next page*

*Figure 9 continued*
The online version of this article includes the following source data and figure supplement(s) for figure 9:

**Source data 1.** Sequencing and mapping QC.

**Source data 2.** Nearest annotated gene with treatment-specific divergence.

**Source data 3.** Nearest annotated gene with punitive de novo singleton mutation.

**Figure supplement 1.** Principal components analysis of single nucleotide polymorphisms that differ between evolved and any ancestral strain (2319 sites).

**Figure supplement 2.** Genome-wide differentiation by (**A**) computed p-values for observed patterns of loss of heterozygosity (LOH) and (**B**) Weir-Cockerham estimator of $F_{ST}$.

Extensive video footage that we recorded of the artificial flowers indicated that the consumption was primarily by hummingbirds.

We used synthetic nectar with either pH = 7, typical of microbe-free fresh nectar, or pH = 2, the lowest potential nectar pH (*Figure 5C*), to evaluate the effect of pH reduction on nectar consumption. In addition, because nectar-colonizing microbes, particularly *M. reukaufii*, can reduce amino acid concentrations in nectar (*Dhami et al., 2016*; *Vannette and Fukami, 2018*), we also manipulated the concentration of amino acids in the artificial nectar to quantify the relative importance of pH compared to amino acid concentration. Nectar contained either a high (0.32 mM) or low (0.032 mM) concentration of amino acids in a factorial design with pH manipulation. We found that for the high amino acid nectar, which resembled the synthetic nectar used in the previously described microcosm experiments, less nectar was consumed from flowers with low-pH nectar (*Figure 10B*). This result

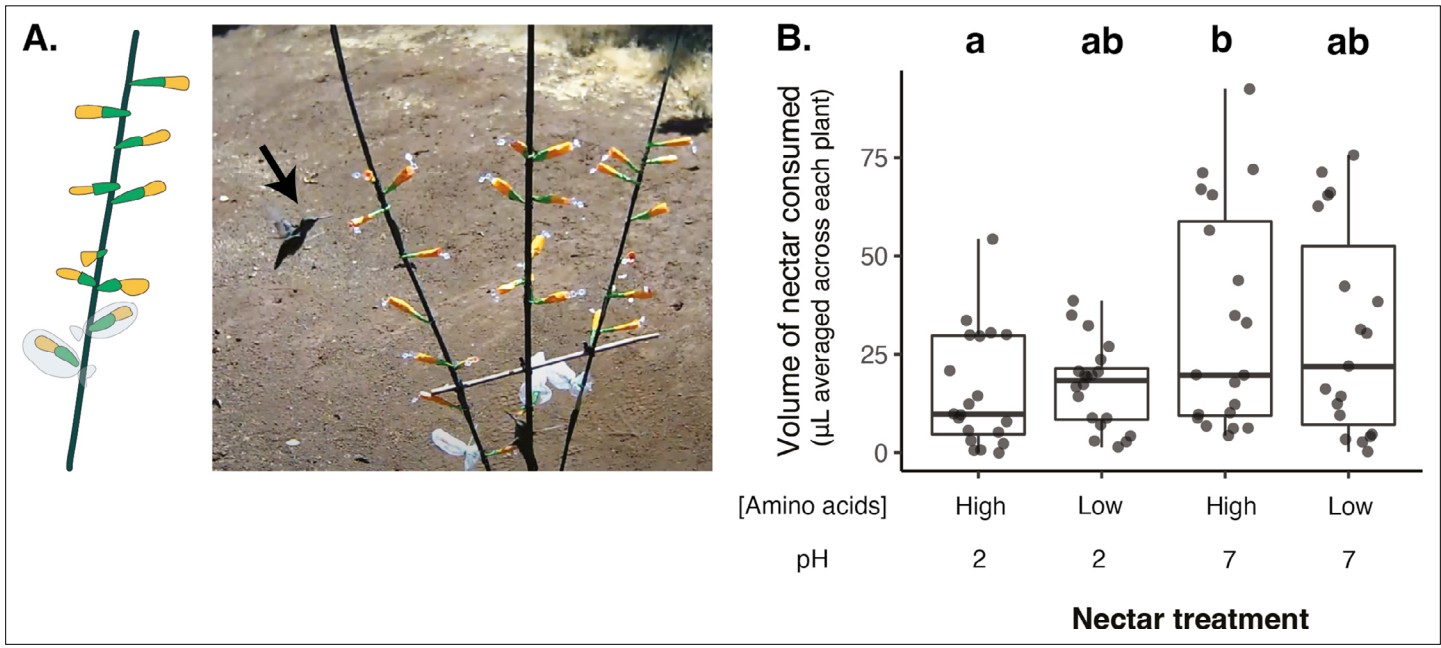

**Figure 10.** Low nectar pH reduces nectar consumption by flower-visiting animals. (**A**) Photo of field experiment setup: hummingbird visits artificial flowers containing PCR tubes of nectar attached to a garden stake. See this link for video: https://www.youtube.com/watch?v=LbD2r43dvnQ or refer to Rich Media (*Figure 10—video 1*) (**B**) Flower-visiting animals consumed less nectar from flowers containing low pH/high amino-acid nectar than high pH/high amino acid nectar (n=79). Letters shown above each box (each treatment) indicate statistical significance as in *Figure 4*.

The online version of this article includes the following video and figure supplement(s) for figure 10:

**Figure supplement 1.** No difference in the volume of nectar consumed by nectar treatment in 2016 (**A**) or 2018 (**B**) when nectar sugars were altered with pH.

**Figure 10—video 1.** Anna's hummingbirds (Calypte anna) visiting experimental plants to consume artificial nectar at the Jasper Ridge Biologial Preserve of Stanford University in California, USA (video taken on July 19th, 2013 by Trevor Hebert and Tadashi Fukami).

https://elifesciences.org/articles/79647/figures#fig10video1

indicates that low pH, but not low concentrations of amino acids, was the primary factor that caused a reduction in nectar consumption. One possible explanation as to why hummingbirds responded less to amino acids than pH is that hummingbirds are less sensitive to the taste of amino acids (*Baldwin et al., 2014*). Because the bacteria lower pH (*Figures 4B and 5C*), these findings suggest that bacteria-dominated flowers can negatively affect nectar consumption by pollinators by lowering nectar pH. We did not observe an effect of pH on pollinator preference when sugar concentrations were also altered, potentially since the pH reduction was less extreme (3.2) in those years or the weather was not as conducive to hummingbird foraging (*Figure 10—figure supplement 1*).

Overall, these results, together with our previous findings (*Vannette et al., 2013*), suggest that in addition to impacting microbial interactions and community assembly, low nectar pH associated with bacteria-dominated flowers affects the relationship between plants and pollinators via microbial priority effects. pH is thought to be a strong predictor of community assembly in other microbial systems as well (*Ratzke and Gore, 2018*), including those in the soil (*Fierer and Jackson, 2006*; *Tedersoo et al., 2014*), the human gut (*Beasley et al., 2015*), and food products such as sourdough bread (*Oshiro et al., 2020*; *Valmorri et al., 2008*), cheese (*Ferreira and Viljoen, 2003*), and milk (*Álvarez-Martín et al., 2008*; *Gadaga et al., 2001*). Priority effects have been observed in some

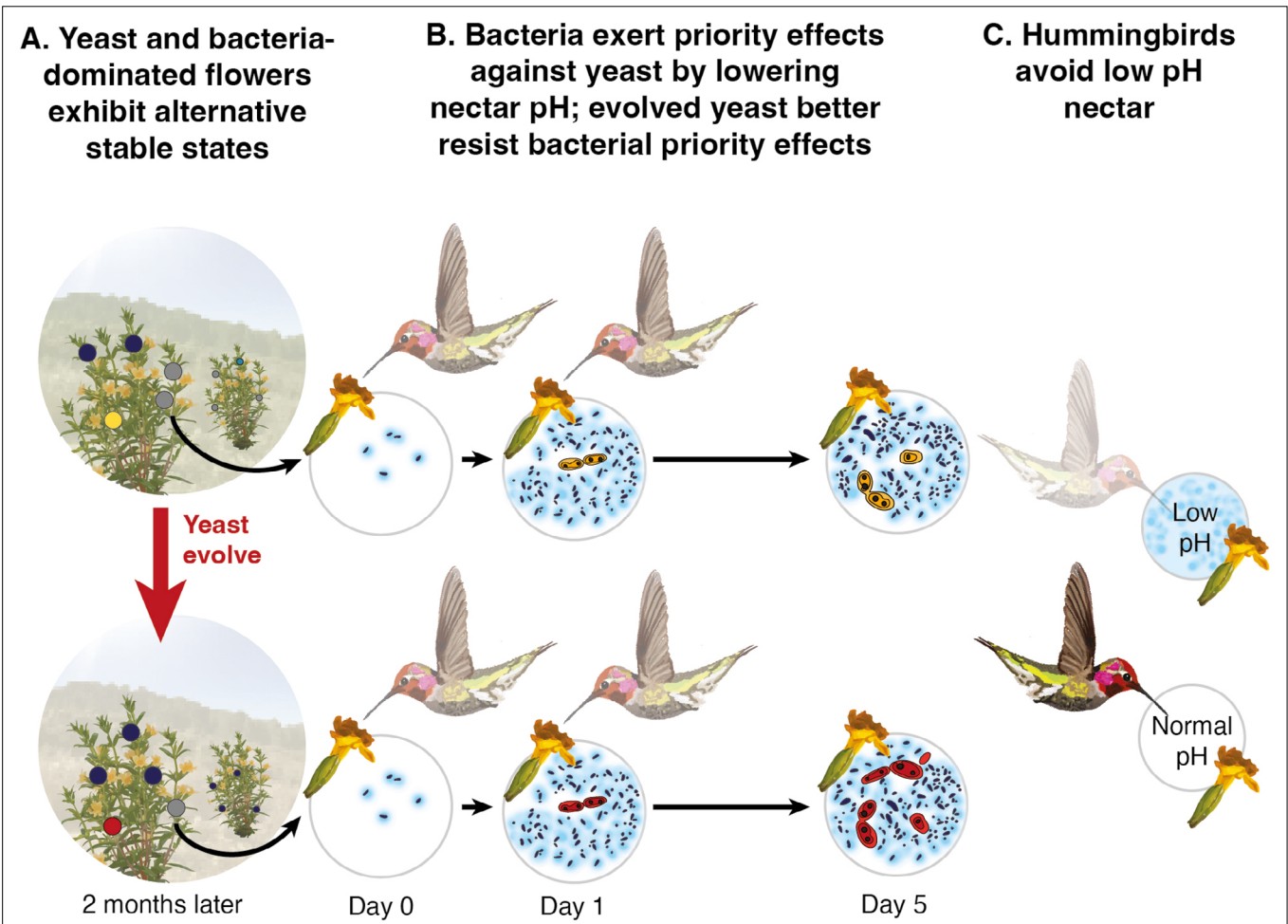

**Figure 11.** pH is an eco-evolutionary driver of priority effects in nectar microbes. (**A**) Field observations show that nectar yeast and bacteria in individual flowers exhibit alternative stable states, where some flowers are either dominated by bacteria (dark blue), yeast (yellow (ancestral) or red (evolved)) or lack significant microbial growth (gray). (**B**) Laboratory experiments identify negative priority effects between bacteria (dark blue) and yeast (yellow (ancestral) or red (evolved)) that lead to alternative states; for example, where early arriving bacteria lower the pH of nectar (depicted here in light blue), limiting the growth of later arriving yeast. Experimentally evolved nectar yeast (red) was less affected by bacterial priority effects, supporting pH as the key mechanism by which nectar bacteria inhibit yeast growth. (**C**) A field experiment shows one functional outcome of bacterial dominance in nectar: low nectar pH decreases nectar consumption by pollinators. Graphics modified from *Chappell and Fukami, 2018*.

of these microbial habitats, suggesting that pH may govern priority effects as an overarching factor across many types of microbial communities.

## Conclusion

The independent series of evidence presented here collectively demonstrates that it is possible to find a single deterministic factor that explains a range of the ecological and evolutionary consequences of priority effects that drive alternative community states (*Figure 11*). Using nectar-inhabiting microbes, we have shown that pH-mediated priority effects (*Figures 4 and 6*) give rise to bacterial and yeast dominance as alternative community states (*Figures 2, 3 and 5*). We have also shown, however, that pH is not merely a mechanism of the priority effects, but it also determines the strength of the priority effects, fuels rapid evolution that changes the strength of the priority effects (*Figures 7–9*), and causes functional consequences of the priority effects for the host plants' reproduction (*Figure 10*). These findings indicate that historical contingency via priority effects can be understood more simply than generally thought once we identify a common factor that dictates various aspects of priority effects. Although this study has focused on one specific system, we hope that its message serves as a source of optimism for evolutionary ecology: even the phenomena regarded as difficult to explain like wide-ranging consequences of historical contingency in community assembly can sometimes be given a surprisingly simple explanation if we study them in depth.

## Materials and methods
### Field survey

Field sampling was conducted at 12 sites throughout the San Francisco Peninsula (*Figure 2*), and a permit was obtained for work at Jasper Ridge Biological Preserve (37°82'40" N, 122°13'30" W). At each site, eight *D. aurantiacus* flowers with a closed stigma were harvested from 12 plants. Flowers with a closed stigma are more likely to have been pollinated than flowers with an open stigma (*Fetscher and Kohn, 1999*). From each flower, nectar was extracted using a clean 10 µL microcapillary tube, measured, and diluted in 40 µL sterile PCR-grade water. Sample size was determined based on previous data (*Vannette et al., 2013*). In the laboratory, samples were further diluted (10 X) and plated on yeast malt agar (YMA; Difco, Sparks, MD, USA) supplemented with 100 mg/L of the antibacterial chloramphenicol and on Reasoner's 2 A agar (R2A; BD Diagnostics, Sparks, MD, USA) supplemented with 20% sucrose and 100 mg/L of the antifungal cycloheximide. Plates were incubated at 25 °C for five days and colony forming units (CFUs) were counted on the YMA and R2A plates to estimate yeast and bacterial abundance, respectively. Molecular identification of colonies via amplification and sequencing of ribosomal genes (16 S rDNA for bacteria and 28 S rDNA for fungi; *Good et al., 2014*) indicated that *Neokomagataea* sp. was capable of forming colonies on the chloramphenicol-supplemented YMA and that these bacterial colonies tended to be distinctly smaller than yeast colonies. Observations of cells from these small colonies under a compound microscope confirmed that the size of the cells tended to be distinctly smaller than cells found in yeast colonies. Furthermore, we found that the small colonies failed to proliferate during subsequent sub-culturing on YMA, whereas most colonies identified as yeast grew optimally, providing additional indication that the small colonies were bacteria. For these reasons, we removed the small colonies from CFU on YMA for our estimation of yeast abundance. In a previous study (*Vannette and Fukami, 2014*), Illumina MiSeq metabarcoding conducted on *D. aurantiacus* nectar samples indicated that the dominant 45 taxa of bacteria (e.g. *Acinetobacter*, *Neokomagataea*, and *Pseudomonas* spp.) and yeast (e.g. *Metschnikowia*, *Starmerella*, and *Cryptococcus* spp.) recovered from culture-independent analyses also form colonies on R2A and YMA, respectively (*Aizenberg-Gershtein et al., 2015*). Thus, our culture-based methods for estimating bacterial and yeast abundance should provide useful, though not perfectly precise, information on their abundances.

We used the CLAM (Classification Methods) program (*Chazdon et al., 2011*), a multinomial model that uses estimated species relative abundance to classify flowers into (1) bacteria-dominated flowers, (2) yeast-dominated flowers, (3) co-dominated flowers, and (4) flowers with too few microbes to be classified into any of the three other groups. The CLAM program was originally developed to classify species into habitat specialists and generalists, focusing on two types of habitats, with species grouped into four categories: habitat A specialists, habitat B specialists, generalists, and species that

are too rare to be classified into any of the other three groups. The same principle can be applied to classifying microbial communities into bacteria-dominated, yeast-dominated, co-dominated, and microbial communities with too few bacteria and/or yeast for classification. We applied the CLAM method separately for flowers in each of the 12 sites. At each site, the resultant flowers (bacteria-dominated, yeast-dominated, co-dominated, too few to classify) were summed and a two-sided Fisher's exact test was used to determine differences between sites.

Culturable fungi and bacteria in *D. aurantiacus* nectar was surveyed at Jasper Ridge Biological Preserve (JR) in 2012, 2016, 2017, 2018, 2019, and 2022. *Diplacus aurantiacus* flowers were chosen haphazardly, and nectar was extracted using microcapillary tubes and diluted into 135 µL sterile 0.85% NaCl. In the lab, nectar was further diluted 10 X and 100 X in sterile 0.85% NaCl. 100 µL of the original diluted nectar was plated on yeast malt agar (YMA; Difco, Sparks, MD, USA) supplemented with 100 mg/L of the antibacterial chloramphenicol, and 100 µL of the 10X-fold diluted nectar was plated onto tryptic soy agar (TSA; BD Diagnostics, Sparks, MD, USA) supplemented with 100 mg/L of the antifungal cycloheximide, or Reasoner's 2 A agar (R2A; BD Diagnostics, Sparks, MD, USA) supplemented with 20% sucrose and 100 mg/L of the antifungal cycloheximide. Plates were incubated at 25 °C for 3–5 days (fungi) or 3 days (bacteria) and the total number of colonies per plate was counted.

Of the plates that had colonies, two colonies of each morphotype (if available) were selected for colony PCR. DNA from individual colonies was extracted by resuspending colonies in 20 µL REDExtract-N-Amp Extraction Buffer (Sigma-Aldrich, Darmstadt, Germany) and lysed at 65 °C for 10 min and 95 °C for 10 min. A total of 20 µL REDExtract-N-Amp Neutralization Buffer (Sigma-Aldrich, Darmstadt, Germany) and 16 µL REDTaq ReadyMix (Sigma-Aldrich, Darmstadt, Germany) were added and samples were amplified via PCR using the NL1/NL4 28 S rDNA primers for fungal DNA samples and E1099FRC/E343F 16 S rDNA for bacterial DNA samples. Amplified samples were treated using 2 µL Exo-SAP (ThermoFisher, Vilnius, Lithuania) and heated for 15 min at 37 °C and 15 min at 80 °C. Successfully amplified samples were sequenced using the NL1 primer for fungal samples and the E343F primer for bacterial samples (*Figure 3—source data 1*).

## Spatial analysis of abiotic factors

To determine whether climatic factors and seasonality influence microbial abundance in this system, WorldClim bioclimatic variables (average annual mean temperature, temperature seasonality, and average monthly precipitation) were extracted for each plant and site. These variables, along with sampling date, were modeled as variables predicting bacterial and fungal abundance, respectively, in a linear mixed model with site as a random effect. Additionally, the geographic distance between plants was regressed against the difference in number of flowers colonized by bacteria or fungi between plant pairs using a linear model. All analyses were conducted using the *glmmTMB* package (1.7.14) in R (3.6.0).

## Nectar pH measurements in the field

*Diplacus aurantiacus* flowers were surveyed for nectar pH measurements at two field sites on the San Francisco Peninsula in May-June of 2022. At Jasper Ridge, 437 flowers were sampled, and at San Gregorio (site 4), 291 flowers were sampled. Flowers were checked for stigma status, where the stigma was identified as either open or closed (*Fetscher and Kohn, 1999*), and anther status, where each flower was categorized from 1 (youngest) to 4 (oldest) as a rough estimate of the flower's age (*Tsuji et al., 2016*). Missing anthers were classified as 0. An anther categorized as a 1 displayed a bright yellow appearance with no brown spots, an anther of 2 was a mix of yellow and golden brown, an anther of 3 was only golden brown, and an anther of 4 was dark brown and wrinkled. Nectar pH was measured using a pH indicator strip (2.0–9.0, EMD, Darmstadt, Germany). Distribution of nectar pH was modeled using an expectation-maximization (EM) algorithm to infer the value of hidden groupings, in this case, three distributions of nectar pH modified by yeast, bacteria, or no microbes. Hartigan's dip test was used to test for multimodality (*dip.test*). Mixture models with k=2 (*Figure 5—figure supplement 1*) and k=3 (*Figure 7*) were generated using the *mixtools* package (1.2.0) Models (k=2 vs. k=3) were compared using two methods. First, a chi-square test was used to compare log-likelihoods between k=2 vs. k=3. Second, AIC values were calculated using both log-likelihoods. All analyses were conducted in R (3.6.0).

Additional nectar sampling to correlate nectar pH and bacterial colony forming units was conducted at Stanford University's Dish Hill from June to August 2022. During this time period, a day prior to sample collection, flowers on *D. aurantiacus* plants that had been planted on Dish Hill in November-December 2021 were bagged to allow for nectar accumulation. After stigma and anther classification, one µL of nectar was extracted using a one µL microcapillary tube, and diluted in 100 µL of sterile 0.85% NaCl. The pH of the remaining nectar was measured using a pH indicator strip (range: 2.0–9.0; EMD, Darmstadt, Germany). After transportation on ice, 10 µL of the diluted nectar was further diluted in 90 µL of sterile 0.85% NaCl (dilution 1) and 10 µL of this diluted sample was further diluted in 90 µL of sterile 0.85% NaCl (dilution 2). 40 µL of dilution 1 was plated onto a yeast malt agar plate (YMA; Difco, Sparks, MD, USA) supplemented with 100 mg/L of the antibacterial chloramphenicol. Forty µL of dilution 2 was plated onto each of the following two media: tryptic soy agar (TSA; BD Diagnostics, Sparks, MD, USA) supplemented with 100 mg/L of the antifungal cycloheximide, and Reasoner's 2 A agar (R2A; BD Diagnostics, Sparks, MD, USA) supplemented with 20% sucrose and 100 mg/L of the antifungal cycloheximide. YMA and TSA plates were incubated at 25 °C and colony forming units (CFUs) were counted after five (YMA and TSA) or three (R2A) days of incubation. YMA plates yielded relatively few CFUs. On TSA, CFUs were counted separately by two easily distinguishable morpho-types (white and yellow colonies). Similarly, on R2A, CFUs were counted separately for white, yellow, and clear colonies. A linear mixed model was used to predict the number of CFUs growing on R2A from nectar pH, with the date sampled and sub-site within the Dish Hill as nested random effects using the *glmmTMB* package (1.7.14) in R (3.6.0).

## Synthetic nectar preparation

Synthetic nectars for laboratory experiments were prepared based on previous chemical analysis of *D. aurantiacus* nectar (*Peay et al., 2012*). The sugars and amino acids found to be abundant in this analysis, including fructose (4%), glucose (2%), sucrose (20%), serine (0.102 mM), glycine (0.097 mM), proline (0.038 mM), glutamate (0.035 mM), aspartic acid (0.026 mM), GABA1 (0.023 mM), and alanine (0.021 mM), were mixed until dissolved, adjusted using NaOH to pH = 6, filtered through a 0.2 µm filter, and stored at –20 °C until used. We made two additional synthetic nectars that were modified from the synthetic nectar recipe above to mimic chemical changes to nectar induced by early arrival of bacteria *A. nectaris*. "Bacteria-conditioned" nectar was prepared by inoculating *A. nectaris* (strain FNA3) at a density of 200 cells/µL into two liters of synthetic nectar. The nectar was conditioned for five days by incubating at 25 °C while shaking at 200 RPM. This conditioned nectar was sterilized by filtering through a 0.22 µm filter and stored at –20 °C until use. This nectar had a pH of 3. The second type of the two additional nectars was 'low-pH nectar', which was prepared by adjusting the pH of the original synthetic nectar to pH = 3 using HCl.

## Experimental evolution

The bacterium *A. nectaris* (strain FNA17, isolated in 2017) and the yeast *M. reukaufii* (strain MR1, isolated in 2010) (Appendix 2) were isolated from floral nectar of *D. aurantiacus* growing at Jasper Ridge Biological Preserve and glycerol stocks were kept at –80 °C. *Acinetobacter nectaris* was streaked on tryptic soy agar (TSA) with 100 mg/L cycloheximide and *M. reukaufii* was streaked on yeast mold agar (YMA) with 100 mg/L chloramphenicol from glycerol stocks. Plates were incubated for two days at 25 °C. Single colonies from these strains were re-suspended in synthetic nectar and inoculated into 96-well plates containing 120 µL nectar per well at a density of approximately 200 cells/µL. Four replicate isolates of each strain were inoculated into low-pH (pH = 3), normal (pH = 6), and bacteria-conditioned (pH = 3) nectar. Plates were covered with sterile, air-permeable membranes (Breathe-Easy sealing membranes, Millipore Sigma, Darmstadt, Germany) and incubated at 25 °C. 10 µL of each culture was transferred to a new 110 µL of nectar for 30 transfers (every Monday, Wednesday, and Friday). Every two weeks, the remaining culture of the evolving strains was frozen in 25% glycerol and stored at –80 °C for later experimental use.

To isolate individual clones of yeast evolved in each treatment, 20 µl of culture from the final transfer was diluted with 200 µl sterile phosphate buffered saline and plated onto YMA plates with 100 mg/L chloramphenicol and incubated at 25 °C for 2 days. From each plate, 10 colonies were re-streaked and stored for future use as single colony pure cultures. One of these isolates per replicate per treatment was used for subsequent priority effects experiments.

## Priority effects experiments

For the first round of priority effects experiments, *A. nectaris* and different strains of *M. reukaufii* (strains MR1, MY0182, and MY0202) were plated on TSA plates with 100 mg/L cycloheximide or YMA plates with 100 mg/L chloramphenicol and incubated for 2 days at 25 °C. Individual colonies from strains were resuspended in synthetic nectar and inoculated into PCR tubes at a density of 200 cells/µL in a fully factorial design (*Figure 4—source data 1*) on either day 0 or day 2 of the experiment. Yeast density was measured using a bright-line hemocytometer (Hausser Scientific, Horsham, PA). Bacterial density was measured using optical density measurements using a microplate reader (TECAN, Männedorf, Switzerland). Microcosms were covered with sterile, air-permeable membranes (Breathe-Easy Sealing Membrane, Sigma-Aldrich, Darmstadt, Germany) and incubated at 25 °C. After 5 days, cultures were resuspended, serially diluted in sterile 0.85% NaCl and plated on TSA plates with 100 mg/L cycloheximide or YMA plates with 100 mg/L chloramphenicol. After two days of incubation at 25 °C, colony forming units were counted to quantify microbial growth in the competition experiment. Experiments were repeated over multiple weeks (rounds) with two or three biological replicates per week.

For the second round of priority effects experiments, ancestral *A. nectaris* (strain FNA17) and ancestral and evolved *M. reukaufii* (strain MR1) were plated on TSA plates with 100 mg/L cycloheximide or YMA plates with 100 mg/L chloramphenicol and incubated for 2 days at 25 °C. Single strains sub-cultured from individual colonies were resuspended in synthetic nectar and inoculated into wells in 96-well plates at either a density of 10,000 cells/µL (a high density of microbes, simulating microbes that arrive early to a flower and subsequently grow to a high density) or 10 cells/µL (a low density of microbes, simulating a late arrival to the flower) with a competitor species or alone in a fully factorial design (*Figure 7—source data 1*). Yeast density was measured using a bright-line hemocytometer (Hausser Scientific, Horsham, PA). Bacterial density was measured using an Attune NxT Flow Cytometer (Invitrogen, Waltham, MA) with Attune autosampler and all data was analyzed using Attune NxT Software (v.3.2.1). The concentration of cells per µL was determined by gating by forward scatter height (FSC) and side scatter height (SSC) at 350 V and threshold of 2.0. After inoculation, 96-well plates were covered with sterile, air-permeable membranes and incubated at 25 °C. After 2 days, cultures were resuspended, serially diluted in sterile 0.85% NaCl, and plated on TSA plates with 100 mg/L cycloheximide or YMA plates with 100 mg/L chloramphenicol. After 2 days of incubation at 25 °C, colony-forming units were counted to quantify microbial growth in the competition experiment. Experiments were repeated over four rounds with two independent replicates (96-well plates) per week. The data from the first week of each academic quarter was omitted from the final analysis due to variation in training new student researchers. Two additional rounds of this experiment were conducted using low-pH nectar instead of the normal nectar to investigate differential growth of ancestral *M. reukaufii* in low- vs. normal-pH nectar. For both experiments, sample size was determined based on previous data (*Belisle et al., 2012*; *Peay et al., 2012*; *Tucker and Fukami, 2014*) and feasibility.

## Priority effects statistical analysis

For all analysis, colony counts were $\log_{10}$-transformed and analysis was conducted using the *glmmTMB* package (1.7.14) in R (3.6.0).

To calculate the effect of arrival order on bacteria and yeast growth, we used a linear mixed model predicting the final density of yeast or bacteria based on the priority effects treatment. We included the week of the experiment (round) and replicate tubes as nested random effects. Then, we used a post-hoc test using the *emmeans* package (1.7.2) to see whether for each priority effect treatment (representing different arrival orders), there was a difference in microbial growth.

To calculate the effect of bacterial density on final nectar pH, we used a Spearman's rank order correlation model with the log-transformed final bacterial density as the predictor and final nectar pH as the response variable.

We calculated the differential response of *M. reukaufii* strains to bacterial priority effects by first calculating priority effect strength (PE) as $\mathrm{PE} = \log\left(\frac{\mathrm{BY}}{-\mathrm{Y}}\right)\ \log\left(\frac{\mathrm{YB}}{\mathrm{Y}-}\right)$, where BY and YB represents early arrival (day 0 vs. day 2 of the experiment) by bacteria or yeast, respectively. -Y and Y- represent the comparable growth of yeast at either arrival time alone (*Vannette and Fukami, 2014*), and treatment densities were averaged by round of the experiment. Next, we used a linear mixed model

predicting the priority effect strength of bacteria on yeast, by yeast strain. We included the week of the experiment (round) and replicate tubes as nested random effects. Then, we used a post-hoc test using the *emmeans* package (1.7.2) to see whether for each yeast strain, there was a difference in the strength of priority effects by bacteria.

For experiments comparing the effect of evolutionary history on priority effect strength (PE), we calculated the priority effect strength (PE), as described above, as well as differences in growth by evolutionary history (ancestral or experimental evolutionary treatment) by subtracting yeast growth with initial bacterial dominance (BY) from monoculture growth at the same density (-Y) for each replicate. We used linear mixed models predicting either the final density of yeast or the difference in yeast growth based on priority effects treatment. We included the week of the experiment (round) and 96-well plate as nested random effects and evolutionary replicate (independent evolutionary trajectory) as a random effect. Then, we used a pairwise post-hoc test using the *emmeans* package (1.7.2) to see whether, for each evolution treatment (ancestral, normal, low-pH, bacteria-conditioned), there was a difference in microbial growth by priority effects treatment (representing different arrival orders).

We calculated the effect of nectar type on final yeast density by conducting the evolutionary priority effects experiments described above in both normal and low pH nectars. By combining these experiments, conducted over 6 total weeks, we used a linear mixed model with the evolution treatment (ancestral or three evolved treatments), the priority effect treatment (related to arrival order), and nectar type as predictors and the final density of yeast as a response. We included independent evolutionary replicates as a random effect and included nested random effects of the week the experiment was conducted and the 96-well plate (experiments were conducted across two 96-well plates). We used a post-hoc test to see whether, for each evolution treatment (ancestral, normal, low-pH, bacteria-conditioned), there was a difference in microbial growth in either low or high pH by priority effects treatment (yeast at a high density, yeast at a low density, etc.).

The relationship between monoculture growth and resistance to priority effects was calculated using a linear mixed model predicting the difference in the strength of priority effects from the monoculture growth. Both values were comparing each evolved strain to the ancestral strain within a single weekly iteration of the experiment, which was also included as a random effect. An additional analysis was conducted comparing the strength of priority effects resistance to monoculture growth without comparison to the ancestral strain (*Figure 8—figure supplement 1*). This analysis was conducted using the *glmmTMB* package (1.7.14) in R (3.6.0).

## Genome re-sequencing

Ancestral *A. nectaris* (strain FNA17) and ancestral and single clones of evolved *M. reukaufii* (strain MR1) from glycerol stocks were plated on TSA plates with 100 mg/L cycloheximide or YMA plates with 100 mg/L chloramphenicol, respectively and incubated for 2 days at 25 °C. Single colonies were selected and subcultured in 10 mL yeast mold broth (yeast) or tryptic soy broth (bacteria). Cultures were grown for 15 hr at 24 °C and 200 RPM. Overnight cultures were adjusted to $1\times10^7$ yeast cells/μL or $1\times10^7$ bacteria cells/μL and yeast cells were treated with Zymolase (100 U/μL) for 30 min at 30 °C. DNA was extracted using Qiagen Blood and Tissue Kit (Qiagen, Redwood City, CA) and quantified using a Qubit HS Kit (ThermoFisher Scientific, Waltham, MA). Dual-indexed genomic libraries were prepared using a Nextera XT index kit (Illumina, San Diego, CA) and pooled for sequencing. Libraries were sequenced on a single Illumina MiSeq V3 run, which produces 300 bp, paired-end reads. This resulted in a total of 675 M mapped reads with an average sequencing depth of 444 X per sample (19.2 Mb genome) (*Dhami et al., 2016*).

## Read filtering, sequence alignment, and variant calling

Variants in genomic data were identified with *grenepipe* (0.1.0) (*Czech and Exposito-Alonso, 2021*), an automated variant calling pipeline for *snakemake* (*Köster and Rahmann, 2012*). Raw fastq sequencing files were trimmed using *trimmomatic* (0.36) (*Bolger et al., 2014*). Filtered reads were mapped to the reference genome provided by *Dhami et al., 2016* using *bwa mem* (0.7.17) (*Li and Durbin, 2009*). Mapped reads were sorted and indexed using *samtools* (1.9) (*Li et al., 2009*) and duplicate reads were removed using the *picard* software (2.22.1) (*Picard Toolkit, 2019*). Genetic variants, both single nucleotide polymorphisms (SNPs) and insertions/deletions (indels) were identified

using *freebayes* (1.3.1) (*Garrison and Marth, 2012*). The resulting VCF file was filtered using *vcftools* (2.22.1) (*Danecek et al., 2011*), which yielded a total of 3,016,402 polymorphic sites across all samples. Quality of filtered reads and their adapter content was determined by *fastqc* (0.11.9) with additional statistics collected from *qualimap* (2.2.2 a) (*Okonechnikov et al., 2016*) for genome coverage, and *samtools stats v1.6* (*Li et al., 2009*) and *samtools flagstat* (1.10) (*Li et al., 2009*) for mapping and alignment metrics. Output of all tools was then summarized in a *MultiQC* (1.9) report (*Ewels et al., 2016*), which we provide as *Figure 9—source data 1*.

## Loss of heterozygosity and de novo mutation analyses

To identify loss of heterozygosity (LOH) between the ancestral strain and the evolved clones, we first identified those sites in which the ancestral strain was heterozygous (26,490 locations, corresponding to ~0.14% of all sites). LOH events were then characterized as those instances in which any of the evolved strains was homozygous at a site identified as heterozygous in the ancestor, an event occurring 1,699 times across all end point clones. We did not expect to see contiguous segments of the genome where all sites show LOH because we do not see recombination in this yeast species (they are clonal), assessed four independent evolutionary replicates per treatment, and conducted the experiment over a relatively short period of time. To explore evidence of treatment-specific LOH events, we computed an odd's ratio of alternate to reference allele counts in the control relative to the low-pH or bacteria-conditioned treatment genotypes. For each odds ratio, we generated 1000 permuted odds ratios in which the genotype data at that site were randomly shuffled across control treatment and either low-pH or bacteria-conditioned samples. P-values were subsequently computed by comparing the observed odds ratio to that of the 1000 permuted ratios.

Next, to further quantify the genomic differentiation between treatments, we computed a Weir-Cockerham estimator of $F_{ST}$ using *vcftools*. While genome-wide $F_{ST}$ was exceedingly low between the normal and low-pH treatment (0.0013), as well as the normal and bacteria-conditioned treatment (0.0014), a small subset of sites (50/26,380) exhibited extreme divergence with $F_{ST}$ ranging between 0.3 and 0.5. Unsurprisingly, those sites with elevated $F_{ST}$ largely overlapped with LOH sites with permutation-derived *P*-values <0.1 (35/25,545). Putative de novo mutations were identified using a custom program to identify SNPs that differed between each pair of ancestral and evolved strains (245 singletons out of 3,016,402 sites). The program was written using the C++library genesis (0.25.0) (*Czech et al., 2020*). To ensure that these singletons were accurate, we generated a mappability score across the genome using GenMap (*Pockrandt et al., 2020*) and filtered by singletons in sites with a mappability score of 1 (representing a unique k-mer region with high mappability), reducing the number of putative de novo mutations to 146 sites.

To explore the functional basis of genomic divergence between ancestral and evolved strains, we identified genes in close proximity to those sites with treatment-specific patterns of LOH (permutation-derived p-value <0.1; $F_{ST}$ = 0.3–0.5) and/or those containing a putative de novo mutation. We re-annotated the *M. reukaufii* reference genome provided by *Dhami et al., 2016* using the InterPro database (*interproscan* 5.45–80.0) (*Blum et al., 2021*) and identified the closest gene with an InterPro annotation within 5 KB upstream/downstream of each candidate site. This annotation resulted in functional information for 42 of the candidate LOH sites and 94 of the putative de novo mutations. Distance to the nearest gene and its associated functional annotation information are provided in *Figure 9—source data 2* and *Figure 9—source data 3*.

## Nectar consumption field experiment

To study the effect of nectar chemistry on nectar consumption by flower-visiting animals, an array of artificial plants was constructed at the Plant Growth Facility on Stock Farm Road on the Stanford University campus. Artificial plants were created using 1 m tall garden stakes with 10 artificial flowers arranged vertically on each stake. Each artificial flower consisted of a pipette tip wrapped in orange and green tape and contained a 200 µL PCR tube of nectar to resemble real *D. aurantiacus* flowers in color, shape, and size. The bottom two artificial flowers were covered with small white organza bags (ULINE, Pleasant Prairie, WI) and were used to estimate the loss of nectar volume by evaporation. See *Figure 10A* for a schematic of these artificial plants. Artificial plants were placed 3–5 m apart and positioned near potted *D. aurantiacus* plants.

Across 2016–2018, 188 artificial plants were deployed, and several nectar types were tested (n=610). In 2017, nectar treatments included two levels of amino acid concentration (high and low) and two pH levels (7 and 2). The high amino acid nectar contained 0.32 mM casamino acids whereas the low amino acid nectar contained 0.032 mM casamino acids. In 2016 and 2018, nectar treatments had two sugar levels (regular and altered sugars) and two pH levels (7 and 3.2). The control sugars had 30% sucrose, 1.5% glucose, and 4% fructose. The altered sugars had 20% sucrose, 0.5% glucose and 6% fructose. Results from these findings are reported in *Figure 10—figure supplement 1*.

Undergraduate students and staff in the BIO 47 (formerly 44Y) course at Stanford University (*Fukami, 2013*) exposed 100 µL nectar in each flower to potential pollinators for 22 hr (approximately 3 PM to 1 PM) and measured the change in nectar using 100 µL Drummond calibrated microcapillary tubes. Nectar removal from each tube was calculated by subtracting the volume of nectar remaining in experimental tubes from the average volume of nectar in bagged controls. Over the duration of the experiment, we observed visitation to artificial flowers by primarily Anna's hummingbirds (*Calypte anna*) and excluded ant visitation by painting the base of each stake with Tanglefoot (Tanglefoot, Marysville, OH, USA). We calculated the difference between treatments and the volume of nectar consumed using a Mann-Whitney U test, and the difference in proportion of flowers consumed using a chi-square test.

## Acknowledgements

We thank Itzel Arias Del Razo, David Cross, Po-Ju Ke, Carolyn Rice, Nic Romano, and Kaoru Tsuji for assistance with field and laboratory work on the field survey (*Figure 2*); Sergio Álvarez-Pérez for his assistance with preliminary priority effects experiments (*Figure 4*); Adrianna Garner, Jonathan Hernandez, and Paloma Vazquez for their assistance with laboratory work with the priority effects microcosm experiments (*Figure 7*); Jonathan Barros and Briana Martin-Villa for assistance with additional analysis; the students, teaching assistants, technical staff, and instructors of the Biology 47 (formerly 44Y) class (*Fukami, 2013*) in 2016–2022, for assistance with the Jasper Ridge field survey (*Figure 3*) and artificial flower experiments (*Figure 10*), as well as in 2019 for assistance with the experimental evolution experiment, in particular Nona Chiariello, Jess Coyle, Bill Gomez, Trevor Hebert, Jesse Miller, and Griselda Morales; Joo Hee Ahn and David Mosko for inspiring us to investigate nectar pH through their Biology 44Y independent project in 2011; Robin Bayer, Hilary Bayer, and other members of the Magic community for assistance with field work at the Dish hill; Leonora Bittleston, Maureen Coleman, Ivana Cvijovic, Moisés Expósito-Alonso, Grant Kinsler, Tadashi Miyashita, Lauren O'Connell, Kabir Peay, Dmitri Petrov, Gavin Sherlock, Meredith Schuman, Rachel Vannette, two anonymous reviewers, and the members of the community ecology group at Stanford for discussion and comments. Funding: This work was supported by the National Science Foundation (DEB 1149600, DEB 1737758), Stanford University's Terman Fellowship, and donation of sequencing materials from Illumina, Inc CRC was supported by a National Science Foundation Graduate Research Fellowship (DGE 1656518) and a Stanford Graduate Fellowship. LC was supported by the Carnegie Institution for Science at Stanford, California, USA. MKD was supported by Marsden Fund Grant (MFP-LCR-2002). SHP was supported by the Life Sciences Research Foundation. Undergraduate researchers F B-B, YC, KE, LG, and CK were supported by the Stanford Department of Biology VPUE Biology Summer Research Program.

## Additional information

### Funding

| Funder | Grant reference number | Author |
|---|---|---|
| National Science Foundation | DEB 1149600 | Tadashi Fukami |
| National Science Foundation | DEB 1737758 | Tadashi Fukami |

| Funder | Grant reference number | Author |
|---|---|---|
| National Science Foundation | DGE 1656518 | Callie R Chappell |
| Marsden Fund | MFP-LCR-2002 | Manpreet K Dhami |
| Life Sciences Research Foundation | | Sur Herrera Paredes |
| Carnegie Institution for Science | | Lucas Czech |

The funders had no role in study design, data collection and interpretation, or the decision to submit the work for publication.

## Author contributions

Callie R Chappell, Conceptualization, Resources, Data curation, Formal analysis, Supervision, Funding acquisition, Validation, Investigation, Visualization, Methodology, Writing – original draft, Project administration, Writing – review and editing; Manpreet K Dhami, Conceptualization, Resources, Data curation, Formal analysis, Supervision, Validation, Investigation, Visualization, Methodology, Project administration, Writing – review and editing; Mark C Bitter, Resources, Data curation, Software, Formal analysis, Supervision, Validation, Investigation, Visualization, Methodology, Writing – review and editing; Lucas Czech, Resources, Data curation, Software, Validation, Investigation, Methodology, Writing – review and editing; Sur Herrera Paredes, Resources, Data curation, Software, Formal analysis, Validation, Investigation, Writing – review and editing; Fatoumata Binta Barrie, Yadira Calderón, Katherine Eritano, Conceptualization, Data curation, Formal analysis, Investigation, Visualization, Methodology, Writing – review and editing; Lexi-Ann Golden, Conceptualization, Data curation, Supervision, Validation, Investigation, Methodology, Writing – review and editing; Daria Hekmat-Scafe, Data curation, Formal analysis, Supervision, Investigation, Methodology, Project administration, Writing – review and editing; Veronica Hsu, Conceptualization, Data curation, Formal analysis, Validation, Visualization, Methodology, Writing – review and editing; Clara Kieschnick, Conceptualization, Data curation, Formal analysis, Validation, Investigation, Visualization, Methodology, Writing – review and editing; Shyamala Malladi, Data curation, Formal analysis, Supervision, Validation, Investigation, Methodology, Project administration, Writing – review and editing; Nicole Rush, Conceptualization, Data curation, Supervision, Validation, Investigation, Methodology, Project administration, Writing – review and editing; Tadashi Fukami, Conceptualization, Resources, Formal analysis, Supervision, Funding acquisition, Investigation, Visualization, Methodology, Writing – original draft, Writing – review and editing

## Author ORCIDs

Callie R Chappell ⓘ http://orcid.org/0000-0003-4611-0021
Mark C Bitter ⓘ http://orcid.org/0000-0001-7607-2375
Lucas Czech ⓘ http://orcid.org/0000-0002-1340-9644
Tadashi Fukami ⓘ http://orcid.org/0000-0001-5654-4785

## Decision letter and Author response

Decision letter https://doi.org/10.7554/eLife.79647.sa1
Author response https://doi.org/10.7554/eLife.79647.sa2

# Additional files

## Supplementary files

• Supplementary file 1. Abstract in German, Hindi, Japanese, Mandarin, and Spanish. Translation of the English abstract in some of the other languages spoken by the authors.

• MDAR checklist

## Data availability

Raw sequencing reads are available at NCBI Sequence Read Archive (BioProject PRJNA825574). All other data and code reported in this paper are

available at: https://gitlab.com/teamnectarmicrobe/n06_nectarmicrobes_ecoevo, (copy archived at swh:1:rev:023e6c17cafa08d701e56dbc9415dfdb75122a8c).

The following datasets were generated:

| Author(s) | Year | Dataset title | Dataset URL | Database and Identifier |
|---|---|---|---|---|
| Chappell CR | 2022 | Metschnikowia reukaufii strain:MR1 Raw sequence reads | https://www.ncbi.nlm.nih.gov/bioproject/?term=PRJNA825574 | NCBI BioProject, PRJNA825574 |
| Chappell CR | 2022 | Eco-evo priority effects microcosm experiments | https://gitlab.com/teamnectarmicrobe/n06_nectarmicrobes_ecoevo/-/tree/main/ecoevo_priority_effects | Gitlab repository, ecoevo_priority_effects |
| Chappell CR | 2022 | Nectar microbes field survey of the greater Bay Area of California | https://gitlab.com/teamnectarmicrobe/n06_nectarmicrobes_ecoevo/-/tree/main/field_survey | Gitlab repository, field_survey |
| Chappell CR | 2022 | Evolved M. reukaufii whole genome resequencing | https://gitlab.com/teamnectarmicrobe/n06_nectarmicrobes_ecoevo/-/tree/main/genomics | Gitlab repository, genomics |
| Chappell CR | 2022 | Pollinator nectar choice field experiment | https://gitlab.com/teamnectarmicrobe/n06_nectarmicrobes_ecoevo/-/tree/main/pollinator_field_experiment | Gitlab repository, pollinator_field_experiment |
| Chappell CR | 2022 | Nectar microbes priority effects microcosm experiment | https://gitlab.com/teamnectarmicrobe/n06_nectarmicrobes_ecoevo/-/tree/main/priority_effects | Gitlab repository, priority_effects |
| Chappell CR | 2022 | pH field survey | https://gitlab.com/teamnectarmicrobe/n06_nectarmicrobes_ecoevo/-/tree/main/pH%20survey | Gitlab repository, pH%20 survey |
| Chappell CR | 2022 | BIO 47 field survey | https://gitlab.com/teamnectarmicrobe/n06_nectarmicrobes_ecoevo.git | Gitlab repository, n06_nectarmicrobes_ecoevo |

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

## Appendix 1

We also looked at culturable fungi along with culturable bacteria at Jasper Ridge (*Figure 3*). In this fungal dataset, too, *M. reukaufii* was common, being the second most common species after *Starmerella bombicola*. However, sample size for the fungal data was more than 10 times smaller (n=53 isolates) than the bacterial data (n=619). Furthermore, the Jasper Ridge isolates were always collected early in the flowering season (mid May), whereas the 12-site regional survey we report in our paper was conducted later in the season (late June to mid July). Seasonal changes in fungal species composition in floral nectar have been documented by other studies at Jasper Ridge (*Vannette and Fukami, 2017*) and elsewhere (e.g., *Brysch-Herzberg, 2004*; *Tsuji and Fukami, 2018*). For these reasons, the data from *Dhami et al., 2018*, where *M. reukaufii* was clearly the most dominant species, are both more extensive and more directly relevant to our present study.

## Appendix 2

Evolving microbes were transferred 30 times at a dilution of 1/10 (10 uL of each culture was transferred to a fresh 110 uL of nectar). These 30 transfers occurred over 10 weeks (1680 hours).

To calculate generation time, we estimated the population size of yeast across the experiment by inoculating a series of nectar microcosms with the same initial density of yeast approximately 2,500 colony forming units (CFUs) and destructively sampling each day.

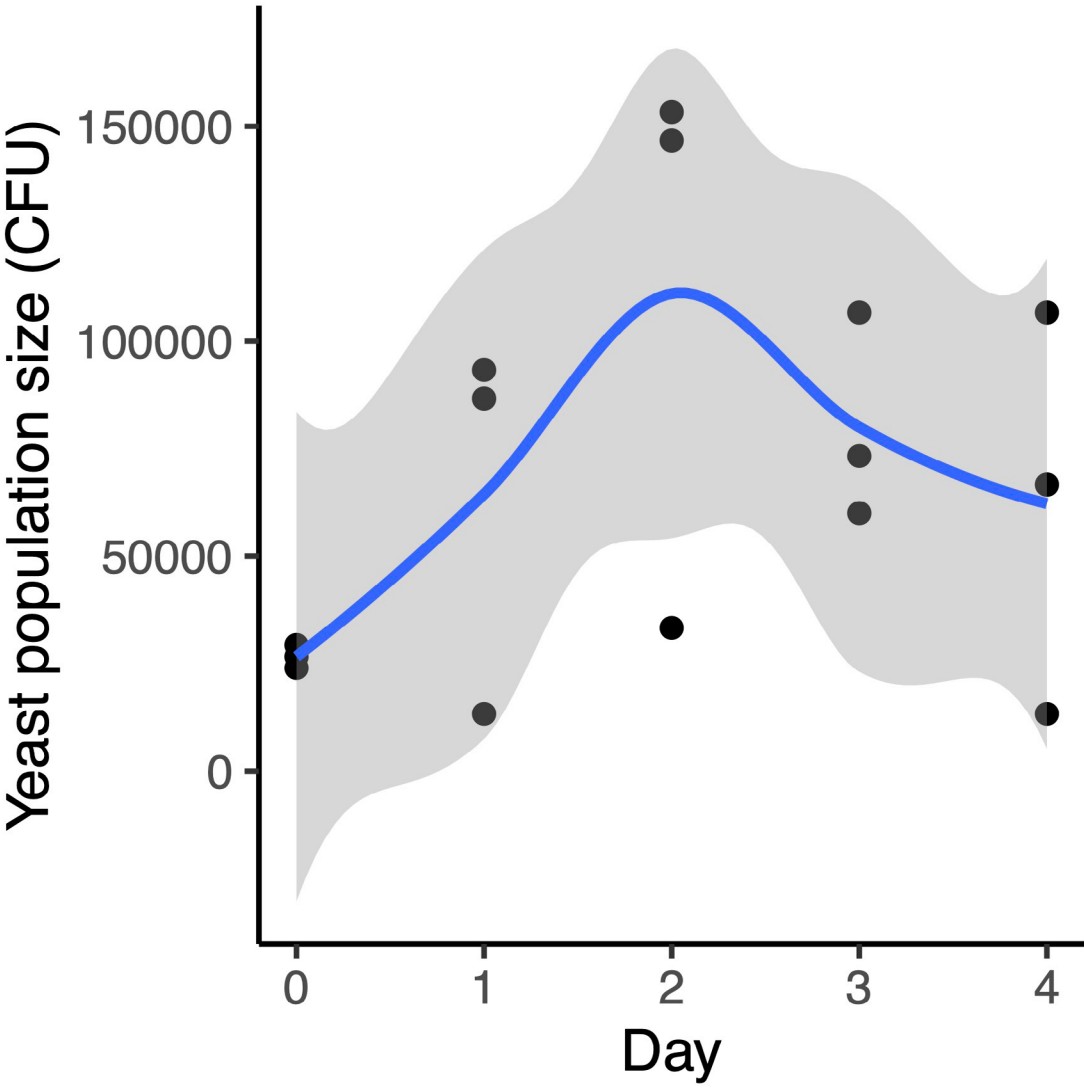

**Appendix 2—figure 1.** M. reukaufii growth over four days in experimental microcosms.

To calculate generation time, we used the following equation, assuming the yeast growth was exponential before day two:

$$\text{Generation time} = \frac{\text{Time (hours)}}{3.3 * \log\left(\frac{\text{Population size of yeast on day two}}{\text{Population size of yeast on day one}}\right)}$$

Based on this equation and the data above, we estimate the generation time of *M. reukaufii* yeast (strain MR1) in synthetic nectar is approximately one generation per eight hours. Assuming the generation time stays constant throughout the experiment, we estimate approximately 200 generations over the 30 transfers.

