## [Editor Report]

This important study documents the causes and consequences of priority effects in community assembly, using a plant-microbe-pollinator model system. Using an elegant combination of lab and field approaches, the authors provide compelling evidence that early-colonizing microbes alter the nectar pH, which has far-reaching ecological and evolutionary effects. It will likely be of interest to a wide audience and has implications for microbial, plant, and animal ecology and evolution.

---

## [Decision Letter]

**Decision letter after peer review:**

Thank you for submitting your article "pH as an eco-evolutionary driver of priority effects" for consideration by *eLife*. Your article has been reviewed by 3 peer reviewers, one of whom is a member of our Board of Reviewing Editors, and the evaluation has been overseen by a Reviewing Editor and Meredith Schuman as the Senior Editor. The following individual involved in the review of your submission has agreed to reveal their identity: Leonora S Bittleston (Reviewer #2).

The reviewers have discussed their reviews with one another, and the Reviewing Editor has drafted this to help you prepare a revised submission. Please note that your point-by-point response must address the essential revisions, but does not need to address the individual reviews.

Essential revisions:

The reviewers all agreed that the work was interesting and well done, but had important suggestions for clarity and presentation. In particular:

1) The resequencing of evolved strains: please clarify this section and add appropriate analysis and/or discussion (see reviewer comments)

2) Field survey: please clarify/expand the diversity of bacteria and yeast from the field survey, other factors that might influence alternative states (e.g. spatial structure, seasonality), and your confidence in colony counts as a measure of abundance.

*Reviewer #1 (Recommendations for the authors):*

In addition to the public comments, here I offer some more specific suggestions for improvement for clarity. I think the genome resequencing part of the paper is a bit of a diversion from the main narrative; it is worth considering whether this is necessary for this particular manuscript. Showing that evolved strains are better at surviving low pH seems sufficient. The discussion of various mutations is rather speculative and would be better expanded (with additional experimental confirmation) in a separate study. (e.g. line 363-366, unclear how pH and amino acids are linked)

There are several paragraphs of discussion that could be reduced and/or removed to streamline the paper and reduce speculation (lines 214-219, 302-316). I found the potential regional evolution scenario to be a bit too speculative without additional data.

Methods (e.g. Line 143-145): I think there needs to be more discussion of whether these CFU counts are valid indicators of total cell abundance (16S data is compositional so doesn't tell you abundance). Are there microscopy counts to validate this approach? Additionally, what about other species in this system? The experiments reduce the system to 1 bacterial and 1 yeast species, but presumably, there are others in the nectar. Is it reasonable to ignore these other species? Are they contributing significantly to cell counts in the field survey?

Line 155, 192: "and vice versa" – but the inverse is never really discussed. Perhaps just mention that this manuscript will only focus on the B→Y effect. Is nutrient draw-down by yeast (line 204-207) the proposed mechanism?

Methods for priority effect experiments: I was unclear on the 10 cells/ul vs 10,000 cells/ul design for early/late. I don't quite understand this experimental design. If this is for yeast, what about bacteria? (or vice versa)

One important question is how generalizable this story is. The experiments showing priority effects use the strain MR1 that seems most vulnerable to these effects. Perhaps the bacterial strain used also causes the most dramatic pH drop. It is hard to tell from the paper whether this 2-strain demonstration is really prevalent in nature. The authors argue that this strain level variation underlies the observed variation in Figure 2A, but a bit more clarity on this issue would help. Also for the pollinator experiments, pH 2 was used, which seems to be the extreme end – again, unclear how general.

Line 430-437: this made me think of food fermentations – I think there are lots of nice examples there of B-Y interactions through pH.

Figure 3C, Supp Figure 3&4 all show "difference in final yeast density" on the y-axis, but it seems to mean something different in each case. Clearly explain the axis in each legend. Why is it negative? If it really is initial-final, this seems counterintuitive for how one would show "growth". Does zero indicate no growth?

Line 737-740 and supp figure 9: I nearly missed this, but it seems important! The data seem to show no effect of pH in these experiments. To avoid the appearance of cherry-picking the data, better explain this discrepancy with the 2017 experiments.

*Reviewer #2 (Recommendations for the authors):*

Figures:

Figure 2: A legend with blue, yellow, and green could be added here. Also, the figure description does not say what the inset is in panel A (looks like log10 version but should be described in the text).

Figure 3, panel B: A legend with the circles vs squares and open vs filled or crosshatched would help readers interpret this panel.

Figure 6, panel A: Why is scaffold 4 shown twice, and with scaffold 3 in between?

Figure supplement 6: Listing the treatment ("Bacteria added before yeast" and "Yeast monoculture") as the title of each panel would make this figure clearer for readers

The trade-off in abundance between yeasts vs. bacteria is outlined (in Figure 2 and related text), but how much diversity is there among the bacteria and yeasts in the broad sampling along the CA coast? Can this be estimated either from the culture data (e.g. morphospecies connected to DNA barcode) or previous work? If it can't be estimated, it would still be good to address it more in the text.

Lines 297-298 and Figure Supplement 6B: It looks like the yeast evolved in bacterially-conditioned nectar grows to significantly lower densities in monoculture when compared with the yeast evolved in normal nectar. This suggests that there could be a trade-off in terms of growth in normal vs. bacterially-conditioned nectar which I think is interesting and should be addressed in the text. Also, were these evolved yeasts all grown in low-pH nectar at the end? If so, what did those densities look like, and was it the same pattern that you see with the BY treatment? This would give more insight into the differences between growing in low pH nectar vs growing with high-density bacteria.

Do you see consistent differences in growth/density among the different replicate strains from the evolution experiment? Perhaps you could tie some of the high LOH variation seen among replicates in Figure 6A to functional differences of the replicate strains.

Line 355: Were there more shared differences in LOH between the low-pH and bacteria-conditioned strains vs. the normal nectar-grown strains? From Figure Supplement 8 it looks like that is not the case. The reported analysis does a good job of looking at the shared ways in which low-pH and bacterial conditioning might lead to genetic changes, but it doesn't address if low-pH is acting in the same way as bacterial conditioning. From Figure 6, it looks like the most LOH was happening in the bacteria-conditioned strains. Panel B also shows a really interesting region between scaffolds 2-3 where a lot of putative de novo mutations are happening only in the bacteria-conditioned strains. Your data presents a great opportunity to delve into genetic change driven by an abiotic vs biotic source.

Methods Section starting at 661: It would be helpful to outline here how many strains were sequenced, and how you arrived at the 444X sequencing depth per sample. What is the estimated genome size for the yeast? Did you also sequence A. nectaris? It sounds like yes from the beginning of the section but I don't see it in the results. Overall, this section needs to be clearer.

Line 709: Small discrepancy. Methods say 156 sites of de novo mutations, while results on Line 368 say 146 de novo mutations across all lineages.

The data, code, and other relevant information all appear to be available and easily accessible.

*Reviewer #3 (Recommendations for the authors):*

– Alternative states: expand on deterministic versus stochastic factors influencing alternative stable states in the Discussion section.

– Clam classification: explain in greater detail how this method classifies flowers (# 47)

– Colony forming units: explain confidence in quantifying microbial growth using colony forming units (# 77).

---

## [Author Response]

Essential revisions:The reviewers all agreed that the work was interesting and well done, but had important suggestions for clarity and presentation. In particular:1) The resequencing of evolved strains: please clarify this section and add appropriate analysis and/or discussion (see reviewer comments)

To address these comments, we have expanded the Discussion section that details the resequencing of evolved strains. We added an additional paragraph comparing genomic differences between bacteria-conditioned and low-pH evolved strains, based on our analysis comparing annotated genes proximal to loci with treatment-specific divergence (lines 520-529).

2) Field survey: please clarify/expand the diversity of bacteria and yeast from the field survey, other factors that might influence alternative states (e.g. spatial structure, seasonality), and your confidence in colony counts as a measure of abundance.

We respond to the three points raised here one at a time. First, to clarify/expand the diversity of bacteria and yeast from the field survey, we now refer to the results published in Dhami et al. (2018), in which we reported that *M. reukaufii* was the most commonly found culturable species of fungi in *D. aurantiacus* nectar samples that we had in the regional survey of the 12 sites used in our present study. For reference, please see Table S2 from Dhami et al. (2018), which summarizes the number of colonies (per species) found at each field site.

As for bacteria, the data we have available from Dhami et al. (2018) are unfortunately not extensive enough to be reliable. However, we added to the manuscript new, previously unpublished data from a six-year survey of nectar-inhabiting bacteria cultured from *D. aurantiacus* nectar at one local site, Jasper Ridge. Even though Jasper Ridge is not one of the sites used in our 12-site survey, it is located within the same region where we did the 12-site survey. Jasper Ridge is the site where much of our prior work studying nectar microbes in *D. aurantiacus* was conducted (e.g., Belisle, Peay, and Fukami 2012; Vannette and Fukami 2018; Vannette, Gauthier, and Fukami 2013; Vannette and Fukami 2017). These longitudinal data were collected since 2016 in collaboration with undergraduate students who took a field biology course at Stanford University (Fukami 2013), which we now summarize in a new Figure 3.

The results in this figure support previously published culture-independent (metabarcoding) data on bacterial species composition of *D. aurantiacus* nectar at Jasper Ridge (Vannette and Fukami 2017) and a nearby site (Toju et al. 2018). Specifically, both the culture-dependent and culture-independent investigations indicate that *Acinetobacter* was the most dominant bacterial genus in *D. aurantiacus* nectar.

We also looked at culturable fungi along with culturable bacteria at Jasper Ridge. In this fungal dataset, too, *M. reukaufii* was prevalent, being the second most frequently cultured species after *Starmerella bombicola*. However, sample size for the fungal data was more than ten times smaller (n = 53 isolates identified by Sanger sequencing) than the bacterial data (n = 619). Furthermore, the Jasper Ridge isolates were collected early in the flowering season (in mid May), whereas the 12-site regional survey we report in our paper was conducted later in the season (in late June to mid July). Seasonal changes in fungal species composition in floral nectar have been documented by other studies at Jasper Ridge (Vannette and Fukami 2017) and elsewhere (e.g., Brysch-Herzberg 2004, Tsuji and Fukami 2018). For these reasons, we are hesitant to make any firm conclusion regarding fungi just from this Jasper Ridge dataset. The data from Dhami et al. 2018 (Table S2 above), where *M. reukaufii* was clearly the most dominant species, are both more extensive and more directly relevant to our present study.

Additionally, the most common species within the most common genera of bacteria and yeast observed in our study system, such as *Acinetobacter, Neokomagataea, Pseudomonas,* and *Metschnikowia*, have been indicated to be common nectar specialists in other plants elsewhere as well, based on culture-independent approaches (e.g., Fridman et al. 2012; Warren, Kram, and Theiss 2020).

Taken together, these independent pieces of evidence on the diversity of bacteria and fungi collectively indicate that, given their prevalence, *A. nectaris* and *M. reukaufii* are a reasonable pair of species to focus on as a first step toward understanding the nectar microbial community of *D. auranaticus* in our study landscape. We have presented and discussed these relevant data in lines 216-235.

As for the second point raised in this comment (other factors that might influence alternative states), we added a series of new analyses to assess potential effects of spatial structure, temperature, precipitation, and seasonality on microbial abundance. In these analyses, we did not detect any statistically significant influence of any of the factors considered. This new finding suggests that the patterns that we present in this paper as being consistent with the possibility of bacterial vs. fungal dominance as two alternative states are unlikely to be explained by alternative factors, reinforcing the conclusion we draw from the field survey data. The new analyses are described in lines 183-205, 707-716, and Figure 2—figure supplements 2-4 in the revised manuscript.

We agree that these factors could potentially explain the presented results. Accordingly, we conducted spatial and seasonal analyses of the data, which we detail below and include in two new paragraphs in the manuscript (lines 183-205):

First, to determine whether spatial proximity influenced yeast and bacterial CFUs, we regressed the geographic distance between all possible pairs of plants to the difference in bacterial or fungal abundance between the paired plants. If plant location affected microbial abundance, one should see a positive relationship between distance and the difference in microbial abundance between a given pair of plants: a pair of plants that were more distantly located from each other should be, on average, more different in microbial abundance. Contrary to this expectation, we found no significant relationship between distance and the difference in bacterial colonization (A, p=0.07, R^2^=0.0003) and a small negative association between distance and the difference in fungal colonization (B, p<0.05, R^2^=0.004). Thus, there was no obvious overall spatial pattern in whether flowers were dominated by yeast or bacteria (Figure 2—figure supplement 4).

Next, to determine whether climatic factors or seasonality affected the colonization of bacteria and yeast per plant, we used a linear mixed model predicting the average bacteria and yeast density per plant from average annual temperature, temperature seasonality, and annual precipitation at each site, the date the site was sampled, and the site location and plant as nested random effects. We found that none of these variables were significantly associated with the density of bacteria and yeast in each plant.

To look at seasonality, we also re-ordered Figure 2C, which shows the abundance of bacteria- and yeast-dominated flowers at each site, so that the sites are now listed in order of sampling dates. In this re-ordered figure, there is no obvious trend in the number of flowers dominated by yeast throughout the period sampled (6.23 to 7/9), giving additional indication that seasonality was unlikely to affect the results (Figure 2—figure supplement 3).

Additionally, sampling date does not seem to strongly predict bacterial or fungal density within each flower when plotted as follows (Figure 2—figure supplement 2):

These additional analyses, now included (Figure 2—figure supplements 2-4) and described (lines 183-205) in the manuscript, indicate that the observed microbial distribution patterns are unlikely to have been strongly influenced by spatial proximity, temperature, moisture, or seasonality, reinforcing the possibility that the distribution patterns instead indicate bacterial and yeast dominance as alternative stable states.

On the third point, to explain our confidence in colony counts as a measure of abundance, we present laboratory data correlating cell counts and colony counts. Please see below and Figure 2—figure supplement 1 for detail.

We have revised the text to address the relationship between cell counts and colony counts with nectar microbes. Specifically, we point out that our previous work (Peay *et al.* 2012) established a close correlation between CFUs and cell densities (r^2^ = 0.76) for six species of nectar yeasts isolated from *D. aurantiacus* nectar at Jasper Ridge, including *M. reukaufii*. For the reviewers’ reference, see Figure S2 in Peay et al. 2012.

As for *A. nectaris*, we used a flow cytometric sorting technique to examine the relationship between cell density and CFU (Figure 2—figure supplement 1). This result should be viewed as preliminary given the low level of replication, but this relationship also appears to be linear, as shown below, indicating that colony counts likely reflect true cell abundance of this species in nectar. We have added the plot to the manuscript as Figure 2—figure supplement 1.

It remains uncertain how closely CFU reflects total cell abundance of the entire bacterial and fungal community in nectar. However, a close association is possible and may be even likely given the data above, showing a close correlation between CFU and total cell count for several yeast species and *A. nectaris*, which are indicated by our data to be dominant species in nectar.

We have added the above points in the manuscript (lines 156-157, 829-833).

Reviewer #1 (Recommendations for the authors):In addition to the public comments, here I offer some more specific suggestions for improvement for clarity. I think the genome resequencing part of the paper is a bit of a diversion from the main narrative; it is worth considering whether this is necessary for this particular manuscript. Showing that evolved strains are better at surviving low pH seems sufficient. The discussion of various mutations is rather speculative and would be better expanded (with additional experimental confirmation) in a separate study. (e.g. line 363-366, unclear how pH and amino acids are linked)

We appreciate this feedback and agree that this part of the study would be strengthened with additional experimental confirmation in a separate study. The phenotypic differences between evolved strains suggest evolutionary change, but we believe it is important to present the genomic resequencing results in this paper as evidence for genetic adaptation as opposed to just plastic and/or epigenetic response. As the reviewer points out, detailed discussion of various mutations needs to be speculative at this stage, however, and we have sought to be clear about it in the text (lines 503-506, 517-519, 520-529). The main point we would like to highlight with the genomic data is that yeast strains responded evolutionarily to discuss eco-evolutionary changes in priority effects. One of the most direct ways to verify if there was evolutionary change is to identify shifts in genetic variation, which is what the genomic results provide. We would also like to point out that reviewer 2 encouraged us to expand this section. To incorporate suggestions from both reviewers, we would like to keep these results in the final manuscript.

There are several paragraphs of discussion that could be reduced and/or removed to streamline the paper and reduce speculation (lines 214-219, 302-316). I found the potential regional evolution scenario to be a bit too speculative without additional data.

We added a line that indicates how they are speculative, “Although we cannot yet directly establish that these are causal variants, they are consistent with our hypothesis that yeasts undergo rapid genetic adaptation to pH-mediated priority effects in nectar” (line 517-519).

Additionally, we clarified line 500 to state, “Competition for amino acids has been suggested as a mechanism by which yeast species exert priority effects against bacteria and other species of yeast in this system (Dhami et al., 2016; Peay et al., 2012; Tucker & Fukami, 2014). Our results suggest that adaptation to limited amino acid availability could be associated with differential resistance to priority effects in low-pH environments, though further experimentation is needed to determine the precise targets of selection.” (lines 500-506).

With respect to the regional evolution scenario, we have added additional analysis showing a tradeoff between local priority effect strength and monoculture growth. See responses to Reviewer 2 for a more detailed discussion of this analysis and how it provides data to support the regional evolution scenario.

Methods (e.g. Line 143-145): I think there needs to be more discussion of whether these CFU counts are valid indicators of total cell abundance (16S data is compositional so doesn't tell you abundance). Are there microscopy counts to validate this approach? Additionally, what about other species in this system? The experiments reduce the system to 1 bacterial and 1 yeast species, but presumably, there are others in the nectar. Is it reasonable to ignore these other species? Are they contributing significantly to cell counts in the field survey?

We have revised the text to address the relationship between cell counts and colony counts with nectar microbes. Specifically, we point out that our previous work (Peay *et al.* 2012) established a close correlation between CFUs and cell densities (r^2^ = 0.76) for six species of nectar yeasts isolated from *D. aurantiacus* nectar at Jasper Ridge, including *M. reukaufii*. For the reviewers’ reference, see Figure S2 in Peay et al. 2012.

As for *A. nectaris*, we used a flow cytometric sorting technique to examine the relationship between cell density and CFU (Figure 2—figure supplement 1). This result should be viewed as preliminary given the low level of replication, but this relationship also appears to be linear, as shown below, indicating that colony counts likely reflect true cell abundance of this species in nectar. We have added the plot to the manuscript as Figure 2—figure supplement 1.

It remains uncertain how closely CFU reflects total cell abundance of the entire bacterial and fungal community in nectar. However, a close association is possible and may be even likely given the data above, showing a close correlation between CFU and total cell count for several yeast species and *A. nectaris*, which are indicated by our data to be dominant species in nectar.

We have added the above points in the manuscript (lines 156-157, 829-833).

Line 155, 192: "and vice versa" – but the inverse is never really discussed. Perhaps just mention that this manuscript will only focus on the B→Y effect. Is nutrient draw-down by yeast (line 204-207) the proposed mechanism?

We would like to point out that, in our original manuscript, we did discuss the inverse priority effects, referring to relevant findings that we previously reported (Tucker and Fukami 2014, Dhami et al. 2016 and 2018, Vannette and Fukami 2018). Specifically, we wrote that: “when yeast arrive first to nectar, they deplete nutrients such as amino acids and limit subsequent bacterial growth, thereby avoiding pH-driven suppression that would happen if bacteria were initially more abundant (Tucker and Fukami 2014; Vannette and Fukami 2018)” (lines 280-283). However, we now realize that this brief mention of the inverse priority effects was not sufficiently linked to our motivation for focusing mainly on the priority effects of bacteria on yeast in the present paper. Accordingly, we added the following sentences: “Since our previous papers sought to elucidate priority effects of early-arriving yeast, here we focus primarily on the other side of the priority effects, where initial dominance of bacteria inhibits yeast growth.” (lines 293-296).

Methods for priority effect experiments: I was unclear on the 10 cells/ul vs 10,000 cells/ul design for early/late. I don't quite understand this experimental design. If this is for yeast, what about bacteria? (or vice versa)

We have added the following explanation to clarify this point: “Single strains sub-cultured from individual colonies were resuspended in synthetic nectar and inoculated into wells in 96-well plates at either a density of 10,000 cells/µL (a high density of microbes, simulating microbes that arrive early to a flower and subsequently grow to a high density) or 10 cells/µL (a low density of microbes, simulating a late arrival to the flower) with a competitor species or alone in a fully factorial design (source data 6). (lines 823-828)”

One important question is how generalizable this story is. The experiments showing priority effects use the strain MR1 that seems most vulnerable to these effects. Perhaps the bacterial strain used also causes the most dramatic pH drop. It is hard to tell from the paper whether this 2-strain demonstration is really prevalent in nature. The authors argue that this strain level variation underlies the observed variation in Figure 2A, but a bit more clarity on this issue would help. Also for the pollinator experiments, pH 2 was used, which seems to be the extreme end – again, unclear how general.

We were also curious whether strain- and species-level variation in the field might complicate our story. To address this point, as well as whether the pH range in the pollinator experiments are realistic, we conducted a new field survey to study the range and distribution of nectar pH in *D. aurantiacus* plants at two sites, Jasper Ridge and San Gregorio (site 5), which we have included as a new paragraph in the manuscript: “To assess whether the nectar pH reduction by bacteria that we observed in the laboratory experiment (Figure 4B) had relevance for explaining variation in nectar pH among real flowers in the field, we conducted a survey of *D. aurantiacus* nectar at two sites, San Gregorio (site 6) and Jasper Ridge, in June and July of 2022. We found that the distribution of nectar pH ranged from 2 to 9 in a way consistent with the prediction that fresh *D. aurantiacus* nectar has a mean pH of about 7.5 and that, once colonized, yeast and bacteria reduce nectar pH to an average of 5.5 and 2.5, respectively (Vannette et al. 2013). We found that the distribution of nectar pH in the field was non-unimodal (Hartigan’s dip test: n=576, D=0.05, p<2.2x10^-16^), and a 3-mode model was a better fit than a 2-mode model (Likelihood ratio test: p=1.6x10^-16^, AIC_k=2_ = 2365, AIC_k=3=_2296, Figure 5—figure supplement 1). According to the mixture model we used, nectar pH had local modes at 7.8, 5.5, and 2.6 (Figure 5A, solid vertical lines), which are strikingly similar to those from the experiment we reported previously (Vannette et al., 2013), where filtered nectar from newly opened *D. aurantiacus* flowers was inoculated with no microbes, the yeast *M. reukaufii*, or the bacterium *Neokomagataea* sp. under controlled laboratory conditions (Figure 5A, dashed vertical lines). In addition, we observed that older and pollinated flowers were more likely to have low nectar pH, reflecting the fact that, in older flowers, microbes would have more chance to grow and modify the nectar pH (Figure 5B, Figure 5—figure supplement 2). We also found that the two sites differed in their distributions of nectar pH (Figure 5B, Figure 5—figure supplement 2). At Jasper Ridge, many of the pollinated flowers had low pH values consistent with bacterial dominance. In contrast, at San Gregorio, these flowers had intermediate pH values that characterize yeast dominance. At another site within the region, Stanford University’s Dish Hill, we measured both nectar pH and bacterial growth and found that *D. aurantiacus* flowers with higher bacterial densities tended to have lower nectar pH (LMM: n=62, p=7.4x10^-8^) (Figure 5C). All of these results give field-based support for the idea that microbial abundance and nectar pH are dictated by the priority effects between bacteria and yeasts” (lines 306-333).

Together, these results support that even when looking at nectar microbe communities in aggregate by looking at a community-level aggregate characteristic like nectar pH, we see differences in community states that could drive the eco-evolutionary processes we observed in our experimental evolution approach.

Line 430-437: this made me think of food fermentations – I think there are lots of nice examples there of B-Y interactions through pH.

We appreciate this suggestion and have added the following to the manuscript: “pH is thought to be a strong predictor of community assembly in other microbial systems as well (Ratzke & Gore, 2018), including those in the soil (Fierer & Jackson, 2006; Tedersoo et al., 2014), the human gut (Beasley et al., 2015), and food products such as sourdough bread (Oshiro et al., 2020; Valmorri et al., 2008), cheese (Ferreira & Viljoen, 2003), and milk (Álvarez-Martín et al., 2008; Gadaga et al., 2001).” (lines 591-596)

Figure 3C, Supp Figure 3&4 all show "difference in final yeast density" on the y-axis, but it seems to mean something different in each case. Clearly explain the axis in each legend. Why is it negative? If it really is initial-final, this seems counterintuitive for how one would show "growth". Does zero indicate no growth?

The Y axis in Figure 3C (now Figure 4C) was incorrect. It should have read “Final yeast density” instead of “Difference in final yeast density,” which we have fixed in the revised manuscript.

For Figure 4—figure supplements 3 and 4, we have reversed the calculation where now, positive and negative values represent population increase and decrease, respectively. This change is also reflected in the figure legends, which now read: “… growth was calculated by subtracting the initial from final cell density. Positive values represent instances of population growth whereas negative values represent instances of population decline.” (lines 1422-1431)

Line 737-740 and supp figure 9: I nearly missed this, but it seems important! The data seem to show no effect of pH in these experiments. To avoid the appearance of cherry-picking the data, better explain this discrepancy with the 2017 experiments.

To address this concern, we have added a line explaining that varying sugars along with pH in 2016 and 2018 may have altered hummingbird visitation and preference to pH during those years. In 2017, the pollinator choice experiment varied pH and amino acid concentration (Figure 10). We found that pH, but not amino acid concentration, affected pollinator preference, consistent with the fact that Anna’s hummingbird (and probably hummingbirds in general) does not respond to amino acid concentration (Baldwin et al. 2014). In 2016 and 2019, we varied pH and sugar concentration (Figure 10—figure supplement 1). It is unclear why we did not detect a pH effect in these years, but one possibility is that pH reduction was less extreme in these years (3.2 in 2016 and 2019, and 2 in 2017). Another possibility is that weather was not as conducive for hummingbird foraging in 2016 and 2019 than in 2017, resulting in less nectar consumption, which may have in turn led to smaller differences among the nectar type treatments. We have added a brief discussion of these possibilities into the manuscript: “We did not observe an effect of pH on pollinator preference when sugar concentrations were also altered, potentially since the pH reduction was less extreme (3.2) in those years or the weather was not as conductive to hummingbird foraging (Figure 10—figure supplement 1)” (lines 582-587).

Reviewer #2 (Recommendations for the authors):Figures:Figure 2: A legend with blue, yellow, and green could be added here. Also, the figure description does not say what the inset is in panel A (looks like log10 version but should be described in the text).Figure 3, panel B: A legend with the circles vs squares and open vs filled or crosshatched would help readers interpret this panel.Figure 6, panel A: Why is scaffold 4 shown twice, and with scaffold 3 in between?Figure supplement 6: Listing the treatment ("Bacteria added before yeast" and "Yeast monoculture") as the title of each panel would make this figure clearer for readers

We have followed all of these suggestions in revised Figure 2, Figure 4, and Figure 6—figure supplement 2. In Figure 9, panel A (formerly Figure 6) the bottom scaffold, formerly listed as scaffold 4 (twice) should have said scaffold 2, which is fixed in the revised figure.

The trade-off in abundance between yeasts vs. bacteria is outlined (in Figure 2 and related text), but how much diversity is there among the bacteria and yeasts in the broad sampling along the CA coast? Can this be estimated either from the culture data (e.g. morphospecies connected to DNA barcode) or previous work? If it can't be estimated, it would still be good to address it more in the text.

We have included two new datasets that address the broader microbial diversity in nectar. First, we have added a new dataset, which concerns similar data collected from *D. aurantiacus* nectar at Jasper Ridge over six years (Figure 3), which gives an idea of the diversity of culturable bacteria and fungi. We have added a paragraph into the manuscript introducing these new data in lines 216-235. Please see response to Essential revision point 2 for further detail.

Second, we now refer to and briefly discuss fungal Sanger sequencing data from a field survey which was previously published (Dhami *et al.* 2018., Table S2) (lines 216-235);

“Previously, we reported that *M. reukaufii* was the most frequently cultured species of fungi in *D. aurantiacus* nectar in the 12-site survey in 2015 (Table S2 in Dhami et al. 2018; see also Belisle et al. 2012). As for bacteria, Dhami et al.’s (2018) data are not extensive enough to draw a firm conclusion. However, we here present data from a multi-year survey of bacteria cultured from *D. aurantiacus* nectar at one site, Jasper Ridge (JR, Figure 3), from 2012 to 2022. Jasper Ridge is not one of the sites used in our 12-site survey, but it is located within the region where we did the survey (Figure 2A). The Jasper Ridge data support culture-independent (metabarcoding) data on bacterial species composition of *D. aurantiacus* nectar at the same site (Vannette and Fukami 2017) and a nearby site (Toju et al. 2018): both culture-dependent and culture-independent methods indicate that *Acinetobacter* spp. were the dominant species of bacteria in D. aurantiacus nectar, followed by *Neokomagataea* (formerly Gluconobacter) sp. (see further detail in Appendix 1). The most common species of bacteria and yeast observed in our study system, such as species of *Acinetobacter*, *Neokamagataea*, *Pseudomonas*, and *Metschnikowia*, have been shown to be common nectar specialists in other plants as well (e.g., Fridman et al. 2012; Warren, Kram, and Theiss 2020). Taken together, these independent pieces of information collectively indicate that, given their prevalence, *A. nectaris* and *M. reukaufii* are a reasonable pair of species to focus on as a first step toward understanding the possible alternative states in the nectar microbial community of *D. aurantiacus* in our study landscape.”

Lines 297-298 and Figure Supplement 6B: It looks like the yeast evolved in bacterially-conditioned nectar grows to significantly lower densities in monoculture when compared with the yeast evolved in normal nectar. This suggests that there could be a trade-off in terms of growth in normal vs. bacterially-conditioned nectar which I think is interesting and should be addressed in the text. Also, were these evolved yeasts all grown in low-pH nectar at the end? If so, what did those densities look like, and was it the same pattern that you see with the BY treatment? This would give more insight into the differences between growing in low pH nectar vs growing with high-density bacteria.

We did additional analysis to determine if there was evidence for the trade-off that the reviewer refers to, and found that there was a not nominally significant (p=0.05) tradeoff between yeasts’ evolved resistance to priority effects and growth in monoculture, when adjusted to growth compared to the ancestral strain. We have included a figure in the main text (Figure 8) and included a brief discussion (lines 428-444):

“Furthermore, we observed a pattern that preliminarily suggests a potential evolutionary tradeoff between adaptation to low pH and neutral pH (Figure 8—Figure 8-source data 1). When considered across all experimental evolution treatments, strains that evolved stronger resistance to priority effects of pH-reducing bacteria became marginally worse at growing in neutral-pH monoculture (strains that are in the upper left quadrant in Figure 8). Conversely, strains that did not increase resistance to bacterial priority effects during experimental evolution were better able to grow in neutral-pH monoculture (those in the upper right quadrant in monoculture) (LMM: n = 48, p = 0.05, Figure 8—figure supplement 1). Incidentally, this type of evolutionary trade-off is the basis for the “eco-evolutionary buffering” hypothesis recently proposed by Wittmann & Fukami (2018). This hypothesis provides one potential reason why species that engage in strong inhibitory priority effects in local communities can still co-exist in a metacommunity even though such priority effects should eventually cause only one species to persist in the metacommunity, with others driven to extinction. Our finding here, i.e., rapid evolution of resistance to priority effects under a trade-off constraint, may partly explain why both yeast and bacteria persist in the field despite strong priority effects, but this possibility remains highly speculative at this stage.”

Unfortunately, we did not grow the evolved yeasts in low-pH nectar in sufficient replication to be reliable for reporting (we did some preliminary trials, but not with adequate replication).

Do you see consistent differences in growth/density among the different replicate strains from the evolution experiment? Perhaps you could tie some of the high LOH variation seen among replicates in Figure 6A to functional differences of the replicate strains.

As shown in the plot in Author response image 1, which is a re-colored version of Figure 7A and indicated by the mixed-effects models we report in the main text, where the evolutionary replicates are incorporated as a random factor, we found no significant difference between evolutionary replicates within each treatment (they were included as a random effect in our linear mixed model). In Author response image 1, each color represents one of the four evolutionary replicates per treatment.

**Author response image 1. sa2fig1:** 

Given that we only measured the final yeast densities with or without bacteria and only four evolutionary replicate strains, we unfortunately do not have enough statistical power to correlate individual polymorphisms with phenotypic differences.

Line 355: Were there more shared differences in LOH between the low-pH and bacteria-conditioned strains vs. the normal nectar-grown strains? From Figure Supplement 8 it looks like that is not the case. The reported analysis does a good job of looking at the shared ways in which low-pH and bacterial conditioning might lead to genetic changes, but it doesn't address if low-pH is acting in the same way as bacterial conditioning. From Figure 6, it looks like the most LOH was happening in the bacteria-conditioned strains. Panel B also shows a really interesting region between scaffolds 2-3 where a lot of putative de novo mutations are happening only in the bacteria-conditioned strains. Your data presents a great opportunity to delve into genetic change driven by an abiotic vs biotic source.

Yeast strains evolved in bacteria-conditioned and low-pH nectar had a similar phenotypic trait, i.e., increased resistance to bacterial priority effects, but we agree that our data suggest that the genetic basis for this trait may have differed. To clarify this point in the manuscript, we have included a more extended discussion speculating on possible differences between two groups of strains.

We agree with the reviewer and have included an expanded discussion in the revised manuscript (lines 520-529). Specifically, to show overall genomic variation between treatments, we calculated genome-wide Fst comparing the various nectar conditions. We found that Fst was 0.0013, 0.0014, and 0.0015 for the low-pH vs. normal, low pH vs. bacteria-conditioned, and bacteria-conditioned vs. normal comparisons, respectively. The similarity between all treatments suggests that the differences between bacteria-conditioned and low pH are comparable to each treatment compared to normal. This result highlights that, although our phenotypic data suggest alterations to pH as the most important factor for this priority effect, it still may be one of many affecting the coevolutionary dynamics of wild yeast in the microbial communities they are part of. In the full community context in which these microbes grow in the field, multi-species interactions, environmental microclimates, etc. likely also play a role in rapid adaptation of these microbes which was not investigated in the current study.

Based on this overall picture, we have included additional discussion focusing on the effect of pH on evolution of stronger resistance to priority effects. We compared genomic differences between bacteria-conditioned and low-pH evolved strains, drawing the reader’s attention to specific differences in Figure 9-source data 2 and 3**.** Loci that varied between the low pH and bacteria-conditioned treatments occurred in genes associated with protein folding, amino acid biosynthesis, and metabolism.

We also looked into the suggested region between scaffolds 2-3 and found that the bacteria-conditioned strains contained putative de novo mutations proximal to genes involved in secretion, oxidative phosphorylation, and membrane insertion in the endoplasmic reticulum. In the future, it would be interesting to study which genes within these regions might be associated with differential responses to each environment.

Methods Section starting at 661: It would be helpful to outline here how many strains were sequenced, and how you arrived at the 444X sequencing depth per sample. What is the estimated genome size for the yeast? Did you also sequence A. nectaris? It sounds like yes from the beginning of the section but I don't see it in the results. Overall, this section needs to be clearer.

The per-sample, genome-wide sequence depth was calculated using MultiQC (the link to the full report is provided in the manuscript). Overall, across-sample average sequencing depth was calculated by taking the average of mean coverage (mean cov) across for each individual sample (n=13). We found fairly even coverage across samples, as seen in Figure 9-source data 1.

However, there is no independent estimate of the genome size of the evolved yeast besides the size of the genome assembly. The mean coverage of the assembled genome is 444X, which might not be exactly the same as 444X sequencing depth, given the evolved yeast genomes could be smaller or larger than the reference genome. Because of the minimal differences between the ancestral and evolved genomes, we expect the numerical difference to be trivial.

We recognize that 444X is unnecessarily high for a genome resequencing project. Our samples were one of the first processed when the sequencing facility reopened after the initial COVID lock-downs in early 2020. Because of the limited number of samples that they had to process, we were allocated more lanes on the sequencer than expected, resulting in the high coverage.

Line 709: Small discrepancy. Methods say 156 sites of de novo mutations, while results on Line 368 say 146 de novo mutations across all lineages.

We corrected this error. It was 146 sites.

The data, code, and other relevant information all appear to be available and easily accessible.Reviewer #3 (Recommendations for the authors):– Alternative states: expand on deterministic versus stochastic factors influencing alternative stable states in the Discussion section.

We have reframed lines 159-181 in the Discussion section to highlight the deterministic versus stochastic factors influencing the alternative stable states observed in our system:

“Across all 12 sites, we found that *D. aurantiacus* flowers were frequently dominated by either bacteria or yeast, but rarely by both (Figure 2C, Figure 2—figure supplement 2). We used a classification method called CLAM (Chazdon et al., 2011) to classify flowers into four groups: bacteria-dominated flowers, yeast-dominated flowers, co-dominated flowers, and flowers with too few microbes to be accurately classified (Figure 2B, Figure 2—figure supplement 3). For each site, we then calculated the proportion of co-dominated flowers that would be expected if bacteria and yeasts were distributed among flowers independently of each other. We found that the observed proportions of co-dominated flowers were lower than expected by chance alone (Figure 2D; paired t-test: n=12, 95% CI [-5.3, -1.1], p=0.006).

This scarcity of co-dominance may have been caused by stochastic factors such as dispersal limitation that creates spatial segregation (Belisle et al., 2012) or deterministic factors such as nectar chemistry that allows for niche partitioning. For example, some flowers may have nectar characteristics that intrinsically favor bacterial growth, while other flowers with different nectar chemistry may preferentially support yeast growth. However, another possibility is that the scarcity of co-dominance was caused jointly by stochastic and deterministic factors. Specifically, if a flower happens to become dominated by bacteria, it may prevent yeast from becoming abundant, and vice versa, through differential modification on nectar chemistry by bacteria vs. yeasts. This mutual suppression would represent inhibitory priority effects, where stochastic dispersal dictates the trajectory of local community assembly because early-arriving colonists deterministically exclude late-arriving immigrants.”

– Clam classification: explain in greater detail how this method classifies flowers (# 47)

We have added more information about the CLAM test into the methods section (lines 669-682):

“We used the CLAM (Classification Methods) program (Chazdon et al., 2011), a multinomial model that uses estimated species relative abundance to classify flowers into (1) bacteria-dominated flowers, (2) yeast-dominated flowers, (3) co-dominated flowers, and (4) flowers with too few microbes to be classified into any of the three other groups. The CLAM program was originally developed to classify species into habitat specialists and generalists, focusing on two types of habitats, with species grouped into four categories: habitat A specialists, habitat B specialists, generalists, and species that are too rare to be classified into any of the other three groups. The same principle can be applied to classifying microbial communities into bacteria-dominated, yeast-dominated, co-dominated, and microbial communities with too few bacteria and/or yeast for classification. We applied the CLAM method separately for flowers in each of the 12 sites. At each site, the resultant flowers (bacteria-dominated, yeast-dominated, co-dominated, too few to classify) were summed and a two-sided Fisher’s exact test was used to determine differences between sites.”

– Colony forming units: explain confidence in quantifying microbial growth using colony forming units (# 77).

We have revised the text to address the relationship between cell counts and colony counts with nectar microbes. Specifically, we point out that our previous work (Peay *et al.* 2012) established a close correlation between CFUs and cell densities (r^2^ = 0.76) for six species of nectar yeasts isolated from *D. aurantiacus* nectar at Jasper Ridge, including *M. reukaufii*. For the reviewers’ reference, see Figure S2 in Peay et al. 2012.

As for *A. nectaris*, we used a flow cytometric sorting technique to examine the relationship between cell density and CFU (Figure 2—figure supplement 1). This result should be viewed as preliminary given the low level of replication, but this relationship also appears to be linear, as shown below, indicating that colony counts likely reflect true cell abundance of this species in nectar. We have added the plot to the manuscript as Figure 2—figure supplement 1.

It remains uncertain how closely CFU reflects total cell abundance of the entire bacterial and fungal community in nectar. However, a close association is possible and may be even likely given the data above, showing a close correlation between CFU and total cell count for several yeast species and *A. nectaris*, which are indicated by our data to be dominant species in nectar.

We have added the above points in the manuscript (lines 156-157, 829-833).